# Molecular mechanism of agonism and inverse agonism in ghrelin receptor

Jiao Qin[1,2,3,4,5,9], Ye Cai[2,9], Zheng Xu[2,9], Qianqian Ming[3,9], Su-Yu Ji[1,3,4,5,9], Chao Wu[2,9], Huibing Zhang[1,3,4,5], Chunyou Mao [6], Dan-Dan Shen [1,3,4,5], Kunio Hirata[7], Yanbin Ma[8✉], Wei Yan [2✉], Yan Zhang [1,3,4,5✉] & Zhenhua Shao [2✉]

Much effort has been invested in the investigation of the structural basis of G protein-coupled receptors (GPCRs) activation. Inverse agonists, which can inhibit GPCRs with constitutive activity, are considered useful therapeutic agents, but the molecular mechanism of such ligands remains insufficiently understood. Here, we report a crystal structure of the ghrelin receptor bound to the inverse agonist PF-05190457 and a cryo-electron microscopy structure of the active ghrelin receptor-Go complex bound to the endogenous agonist ghrelin. Our structures reveal a distinct binding mode of the inverse agonist PF-05190457 in the ghrelin receptor, different from the binding mode of agonists and neutral antagonists. Combining the structural comparisons and cellular function assays, we find that a polar network and a notable hydrophobic cluster are required for receptor activation and constitutive activity. Together, our study provides insights into the detailed mechanism of ghrelin receptor binding to agonists and inverse agonists, and paves the way to design specific ligands targeting ghrelin receptors.

[1] Department of Biophysics and Department of Pathology of Sir Run Run Shaw Hospital, Zhejiang University School of Medicine, 310058 Hangzhou, China. [2] Division of Nephrology and Kidney Research Institute, State Key Laboratory of Biotherapy and Cancer Center, West China Hospital, Sichuan University, 610041 Chengdu, Sichuan, China. [3] Liangzhu Laboratory, Zhejiang University Medical Center, 311121 Hangzhou, China. [4] MOE Frontier Science Center for Brain Research and Brain-Machine Integration, Zhejiang University School of Medicine, 310058 Hangzhou, Zhejiang, China. [5] Zhejiang Provincial Key Laboratory of Immunity and Inflammatory Diseases, 310058 Hangzhou, China. [6] Department of General Surgery, Sir Run Run Shaw Hospital, Zhejiang University School of Medicine, 310016 Hangzhou, China. [7] RIKEN SPring-8 Center, Sayo-cho, Sayo-gun, Hyogo 679-5165, Japan. [8] Institute of innovation, GeneScience Pharmaceutical Co., Ltd., Shanghai, China. [9] These authors contributed equally: Jiao Qin, Ye Cai, Zheng Xu, Qanqian Ming, Su-Yu Ji, Chao Wu. ✉email: mayanbin@gensci-china.com; weiyan2018@scu.edu.cn; zhang_yan@zju.edu.cn; zhenhuashao@scu.edu.cn

An increasing number of GPCRs have been reported to exhibit high basal or constitutive activity in the absence of extracellular stimuli[1–4]. The spontaneous manner of GPCRs is implicated in human physiological functions and various disorders[5]. As a result, inverse agonists have been discovered as a new ligand category alongside agonists and neutral antagonists[6]. In pharmacology, inverse agonists bind to the orthosteric site of receptors and reduce the constitutive activity of receptors, causing action exactly opposite to agonism and resulting in a paradigm shift in the field of GPCR pharmacology[7,8]. Therefore, understanding the constitutive activity of GPCRs and the mechanism of inverse agonism would contribute to the development of new therapeutic drugs.

The ghrelin receptor[9], also known as the growth hormone secretagogue receptor (GHSR), belongs to the β-branch of class A GPCRs and displays constitutive activity[10] (50% activity independent of the endogenous ligand ghrelin). The ghrelin receptor exerts a wide range of physiological functions, including appetite regulation, alcohol consumption, adipocyte metabolism, and glucose homeostasis[11–14], due to its broad distribution and multiple signaling pathways through divergent G-protein coupling or β-arrestin recruitment[15]. The endogenous agonist ghrelin is a 28-amino acid peptide secreted primarily by the stomach, and an acylated modification, typically an octanoyl group, of the hydroxyl group of the $Ser^{+3}$ residue is required for the biological action of ghrelin as the unacylated form of ghrelin does not bind or activate the ghrelin receptor at all[16,17]. In addition, the length of fatty acid modification can also produce diverse activation potency by comparison with the common octanoylated ghrelin in vivo[18]. Previous studies have suggested that inhibition of the ghrelin–ghrelin receptor signaling axis and deacylation of ligands or deletion of receptors could potentially prevent obesity and type 2 diabetes (T2D); thus, blockade of the ghrelin receptor has been proven to be a great therapeutic approach for the treatment of related diseases[19–23].

Given the constitutive activity of the ghrelin receptor, inverse agonists would be the target of pharmacological agents for maximal efficacy[24]. To date, several reported ligands from different pharmaceutical companies display consistent inverse agonism;[25,26] however, in some cases, preclinical research is confounding. PF-05190457 is the only reported small-molecule inverse agonist targeting the ghrelin receptor progressing to phase 1 clinical trial for T2D and alcoholism treatment[27–29]. Despite a previous study revealing a bifurcated pocket in the neutral antagonist-bound ghrelin receptor structure[30], the molecular recognition of PF-05190457 by ghrelin receptors remains unclear, impeding effective drug development.

In this work, we determine the crystal structure of the ghrelin receptor in complex with the inverse agonist PF-05190457 and the cryo-electron microscopy (EM) structure of the active ghrelin receptor bound to endogenous ghrelin coupled to the Go heterotrimer. Our study provides an opportunity to comprehensively understand the distinct conformations of ghrelin receptors bound to different types of ligands. Moreover, our structures reveal a distinct binding mode of inverse agonists with receptors, define cavities for ligand recognition, and decipher the action mechanism of inverse agonists and agonists for ghrelin receptors.

## Results
**Overall structures of the ghrelin receptor in complex with inverse agonist and endogenous agonist.** The ghrelin receptor is activated by endogenous ghrelin peptide and contains constitutive activity as ~50% of the maximal activity in the absence of ghrelin peptide[31]. Antagonist compound 21 had no effect on the constitutive activity of the ghrelin receptor[30], whereas the constitutive

activity of the ghrelin receptor was significantly reduced upon the addition of the inverse agonist PF-05190457 (Fig. 1a).

To describe the molecular mechanism of the ghrelin receptor in binding with PF-05190457, the initial construct of the ghrelin receptor was truncated by eliminating the first N-terminal 34 amino acid residues and the C-terminus after residue 342. To facilitate crystallization, a thermostabilized apocytochrome bRIL was fused to the third intracellular loop (ICL3). Furthermore, mutations of $T130^{3.39}K$ and $N188^{ECL2}Q$ were introduced to improve the thermostability and homogeneity of the receptor as described in our previous study[32] (Supplementary Fig. 1a–d). This construct was crystallized in a complex with PF-05190457 and determined at a resolution of 2.94 Å (Fig. 1b). In the resolved crystal structure, the PF-05190457-bound ghrelin receptor is packed in a $P2_1$ monoclinic lattice with two complex molecules per asymmetric unit (Supplementary Fig. 1e). The two receptor molecules display high identity with an RMSD value of 0.3 Å (root-mean-square deviation of $C_\alpha$) (Supplementary Fig. 1f).

In addition, we determined the ghrelin-bound ghrelin receptor-Go complex using a single-particle cryo-EM technique. The full-length of wild-type human ghrelin receptor, the thermostabilized $miniG\alpha o1^{33}$, Gβ1, and Gγ2 were co-expressed in insect cells. We used the NanoBiT tethering strategy[34] to

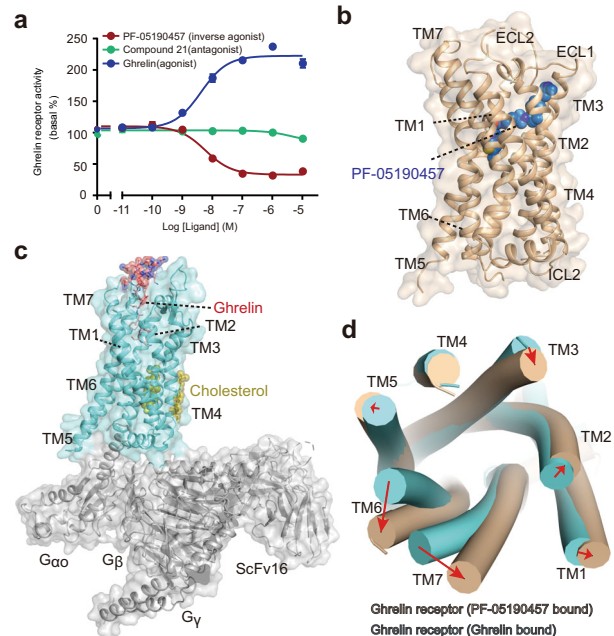

**Fig. 1 Overall structures of ghrelin receptors bound to inverse agonist and agonist. a** Dose-dependent responses of endogenous agonist ghrelin, inverse agonist PF-05190457 and antagonist compound 21 containing 10 nM ghrelin peptide at wild-type ghrelin receptor measured by cellular IP1 accumulation assays. The ghrelin peptide represents high activation potency as a full agonist, and PF-05190457 functions as an inverse agonist to significantly reduce basal activity (dotted line). Data represent the mean ± SEM from $n = 3$ biologically independent experiments performed in triplicate. **b** View from within the plane of the membrane. The inverse agonist-bound ghrelin receptor is represented as a wheat cartoon, and PF-05190457 is shown as a marine blue sphere. Cryo-EM structure of the agonist ghrelin-bound ghrelin receptor-Gαo1-scFv16 complex. **c** Ghrelin peptides and receptors are shown as deep salmon spheres and aquamarine cartoons, respectively. The Go heterotrimer and scFv16 are shown in the gray cartoon. **d** View from the extracellular side of the membrane. The structural comparison of inverse agonist-bound (wheat) with agonist-bound structures (aquamarine) reveals that notable outward movements occur in the extracellular ends of TM domains.

stabilize the complex, in which the C-terminus of ghrelin receptor was attached to LgBiT subunit and $G\beta1$ was fused with a C-terminal HiBiT subunit. An antibody fragment, scFv16, was also added. The final cryo-EM map has a nominal resolution of 2.8 Å after refinement, and a clear density map allows us to build the ghrelin peptide, receptor, Go, $G\beta$, $G\gamma$, and scFv16 (Fig. 1c and Supplementary Figs. 2 and 3). The ghrelin receptor-Go complex displays similar conformations compared with previously solved activated GPCR-Go structures (Supplementary Fig. 4), suggesting that the ghrelin receptor bound to ghrelin was in an activated state.

Compared with the active structure, the extracellular portion of PF-05190457-bound ghrelin receptors, especially TM6 and TM7, displayed notable outward movement, enlarging the orthosteric pocket of the receptor (Fig. 1d). In addition, distinct conformational changes were also found in the cytoplasmic region of the receptor, in which TM6 swing inwards by ~10 Å to hinder the coupling of G proteins (Supplementary Fig. 5a–c).

**Inverse agonist binding pocket of ghrelin receptor**. The compound PF-05190457 was synthesized from a spiro-azetidino-piperidine analog to improve the selectivity and inverse agonism[27]. Like most inverse agonists, the PF-05190457-bound ghrelin receptor complex reflects an inactive state conformation that differs from the complex bound with agonists or antagonists[35–37] (PDB ID: 6CM4, 6K1Q, 5U09). Strikingly, unambiguous electron density at the orthosteric pocket placed PF-05190457 at an unusual site toward TM2, TM3, and TM6 (Fig. 2a), contrasting with the site occupied by the antagonist in the ghrelin receptor[30] (Fig. 2b). The antagonist compound 21-bound ghrelin receptor complex structure demonstrates a bifurcated ligand-binding pocket separated by a salt bridge between $E124^{3.33}$ and $R283^{6.55}$, which is referred to as cavity I and II[30] (Fig. 2b). Whereas PF-05190457 adopts an extended conformation (Fig. 2c–g), the arm-1 ((6-methylpyrimidin-4-yl)-2,3-dihydro-1 H-inden-1-yl moiety) (Fig. 2c) of the ligand projects into the cleft between TM2 and TM3 and is covered by ECL1 and ECL2 regions, making van der Waals contacts and hydrophobic interactions with $R102^{2.63}$, $Q120^{3.29}$, and $F119^{3.28}$ as well as disulfide-bound $C116^{3.25}$ and $C198^{ECL2}$ (Fig. 2d). The cleft between TM2 and TM3 accommodates arm-1 of PF-05190457 is defined as cavity III. Two residues, $F119^{3.28}$A and $Q120^{3.29}$A substitution, were found to be essential for the inverse agonism potency of PF-05190457 (Supplementary Figs. 6a and 7a, b).

The diazaspiro core (2,7-diazaspiro [3,5] nonan-7-yl moiety) of PF-05190457 is adopted in cavity I, forming direct interactions with $D99^{2.60}$ and $S308^{7.38}$ by hydrogen bonding (Fig. 2c, e). Alanine replacement with $D99^{2.60}$ significantly reduced the potency of inverse agonism induced by PF-05190457 (Fig. 2g and Supplementary Fig. 7a, b), indicating that $D99^{2.60}$ may be a key facet for inverse agonist recognition or receptor activation, which is consistent with a previous report that the diazaspiro core moiety was typically required for inverse agonism[27,38]. In addition, mutation of $S308^{7.38}$ to alanine decreased the inverse agonism potency of PF-05190457, which may affect the recognition of the ligand and reduce the stability of the inactive conformation (Fig. 2g and Supplementary Fig. 7a, b). Both the $D99^{2.60}$A and $S308^{7.38}$A mutants retained the expression level of the wild-type ghrelin receptor by normalization (Supplementary Figs. 6a and 7b).

Arm-2 (2-methylimidazo moiety) penetrates deeply into the helical core of the receptor, and it packs against the side chains of residues $F279^{6.51}$, $W276^{6.48}$, and $F312^{7.42}$ in TM6 and TM7, opening another hydrophobic pocket defined as cavity IV (Fig. 2f). In addition, arm-2 is also observed to form hydrogen bonds with residues $W276^{6.48}$ and $S308^{7.38}$ (Fig. 2f). Disrupting the hydrophobic pocket by replacing $F279^{6.51}$, $W276^{6.48}$, and $F312^{7.42}$ with alanine significantly reduced the inverse agonistic activity (Fig. 2g). Moreover, these mutations also impaired the potency of the agonist ghrelin peptide and antagonist compound 21 (Fig. 2g and Supplementary Fig. 6b), indicating that aromatic residues are essential for the activation transition of the ghrelin receptor.

Structural comparison of PF-05190457-bound with compound 21-bound ghrelin receptor complex reveals a notable difference. The salt bridge between $E124^{3.33}$ and $R283^{6.55}$ formed in the antagonist receptor structure is disrupted upon receptor binding with PF-05190457 (Fig. 2h). Residue $R283^{6.55}$ swings away from $E124^{3.33}$, leading to the disappearance of the boundary between cavity I and cavity II. In contrast to the obvious change in the extracellular portion, the cytoplasmic ends of TM5 and TM6 in the PF-05190457-bound structure display conformations similar to those of the inactive ghrelin receptor, suggesting that the inverse agonist stabilizes the receptor in an inactive conformation (Supplementary Fig. 5d–f).

**Agonist binding pocket of ghrelin receptor**. Distinct from the small-molecule inverse agonist binding mode, the endogenous ghrelin peptide is well folded and stably anchored into its binding site through an extensive network of contacts with the receptor. The synthetic ghrelin peptide contains 28 residues, and an octanoyl modification was introduced at the side chain of $Ser^{+3}$ (the superscript $+n$ indicates the amino acid position of ghrelin peptide). The first sixteen residues are clear in our density map (Supplementary Fig. 3). The N-terminal portion of ghrelin from $Gly^{+1}$ to $Pro^{+7}$ penetrates into the conserved orthosteric pocket (Fig. 3a), where $Gly^{+1}$ and $Ser^{+2}$ fill cavity I and are stabilized by a hydrogen-bound network formed by $S123^{3.32}$, $N305^{7.35}$, and $R283^{6.55}$ in the ghrelin receptor (Fig. 3b). Meanwhile, $R283^{6.55}$ is stabilized via a salt bridge with $E124^{3.33}$ and a hydrogen bond with $S217^{5.43}$ (Fig. 3b).

Notably, the interaction between $S217^{5.43}$ and $R283^{6.55}$ was found only in the agonist-bound structure and not in the antagonist-bound structure (Supplementary Fig. 8a). These observations, particularly for the rotamer change of $R283^{6.55}$, reveal that rearrangement of the residues in the polar network appears to tether those key residues on TM3, TM5, TM6, and TM7, contracting the agonist binding pocket of the receptor. In agreement with our structural comparison, alanine substitution of residues $E124^{3.33}$ and $N305^{7.35}$ in the receptor remarkably reduced the activation potency induced by the ghrelin peptide, while alanine substitution of residue $R283^{6.55}$ almost abolished the efficacy (Fig. 3c), suggesting that the polar interactions of $E124^{3.33}$, $R283^{6.55}$, $S217^{5.43}$, and $N305^{7.35}$ in the receptor and $Gly^{+1}$ and $Ser^{+2}$ in the ghrelin peptide play significant roles in receptor activation. In accordance with previously published Gq-coupled ghrelin receptors in complex with ghrelin structure[39], $Gly^{+1}$ and $Ser^{+2}$ in ghrelin are engaged in similar contacts with receptors (Supplementary Fig. 8b).

In the GPCR activation transition, rearrangement of the polar network is required for conformational propagation. The activation process of $\beta2AR$, for example, is the best-characterized member of the GPCR family; residues $D^{3.32}$, $S^{5.42}$, and $S^{5.46}$ from TM3 and TM5 constitute a key polar motif for distinguishing different types of ligands and trigger signaling pathways (PDB: 4LDO)[40], and a polar interaction manner was also found in the D1 receptor (PDB: 7CKZ)[41] (Supplementary Fig. 8d–f). Despite the divergent nature of the endogenous ligand between the ghrelin receptor and aminergic receptors, the ghrelin receptor appears to have a similar activation process as aminergic receptors.

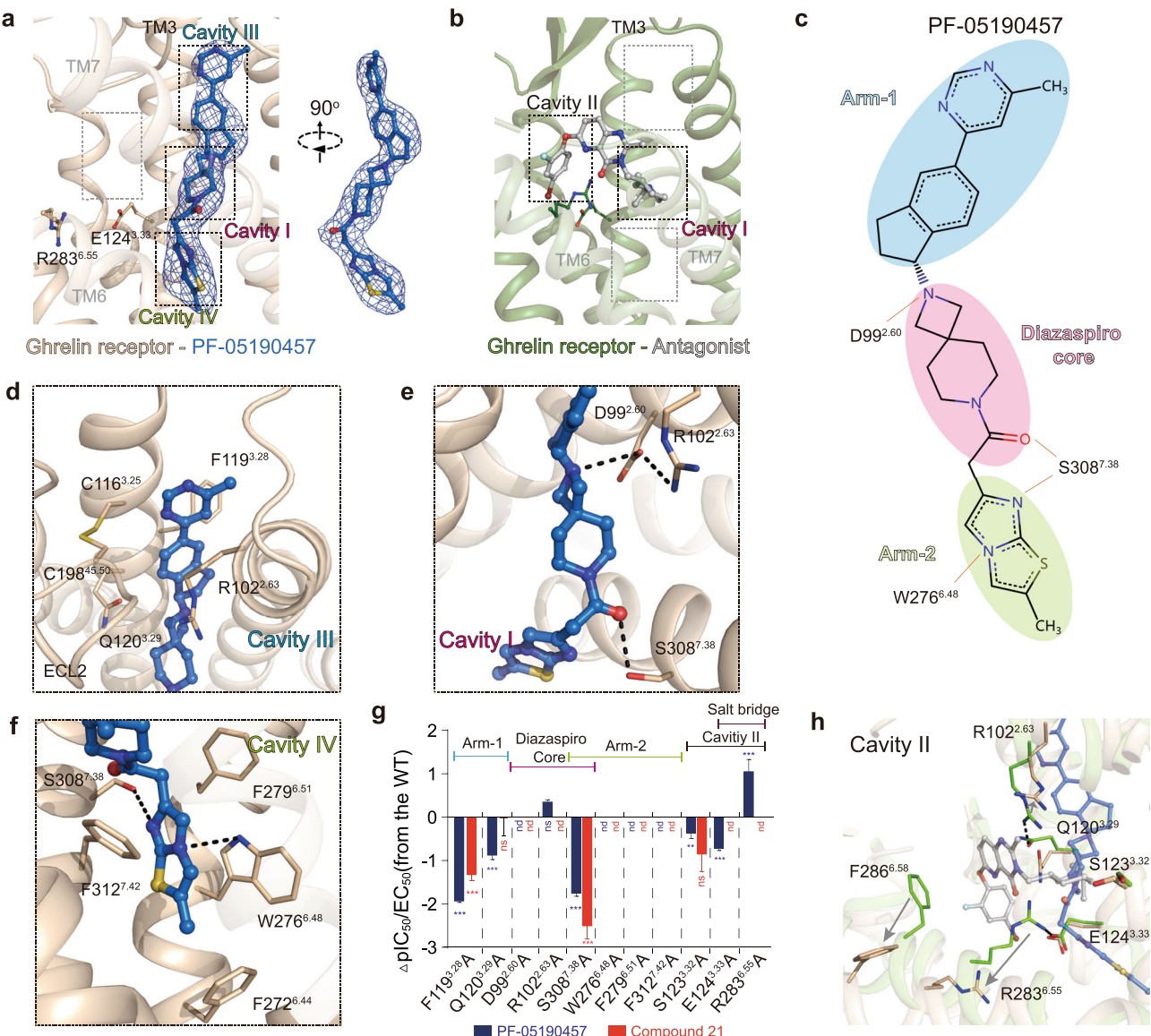

**Fig. 2 The binding mode of PF-05190457 with the ghrelin receptor. a** The detailed binding mode of PF-05190457 in the orthosteric pocket of the ghrelin receptor. The |Fo| − |Fc| omit map (contoured at 3.0 σ) for PF-05190457 (shown as marine blue sticks). In particular, the herein described cavities of the ghrelin receptor named cavity III and cavity IV. **b** The structure of the neutral antagonist (gray stick)-bound ghrelin receptor (forest cartoon) (PDB: 6KO5) reveals a bifurcated ligand-binding pocket, which is referred to as cavities I and II. **c** Two-dimensional structures of the inverse agonist PF-05190457 and the groups of ligands are termed the diazaspiro core, arm-1 and arm-2. The diazaspiro core is shown in pink, arm-1 is sky blue, and arm-2 is light green. The red solid lines indicate hydrogen bonds involved in interactions with the side chains of key residues. **d–f** Key residues of the ghrelin receptor (wheat sticks) involved in ligand binding in cavity III (**d**), cavity I (**e**), and cavity IV (**f**). The binding pocket is highly hydrophobic, and the particular polar network is shown as black dotted lines. **g** IP1 accumulation induced by the inverse agonist PF-05190457 and antagonist compound 21. Bars represent the difference in the calculated potency (pIC$_{50}$) of the inverse agonist for mutations relative to the WT ghrelin receptor. Data are colored according to extent of effect. nd not detected, **$P < 0.01$, ***$P < 0.001$ (one-way analysis of variance (ANOVA) followed by Dunnett's test, compared with the response of WT, $P < 0.001$, $P < 0.001$, $P < 0.001$, $P > 0.999$, nd, nd, $P = 0.05$, nd, $P < 0.001$, $P < 0.001$, nd, nd, nd, nd, nd, nd, $P < 0.01$, $P = 0.264$, $P < 0.001$, nd, $P < 0.001$, nd from left to right). Data represent the mean ± SEM from $n = 3$ biologically independent experiments performed in triplicate. See also Supplementary Figs. 6 and 7. **h** Structural superimposition of inverse agonist-bound with neutral antagonist-bound structures shows significant displacements of the critical residues in the orthosteric pocket. The salt bridges between E124$^{3.33}$ and R283$^{6.55}$ as well as between R102$^{2.63}$ and Q120$^{3.29}$, are broken upon PF-05190457 binding.

Although residue Ser$^{+3}$ of ghrelin could be modified with different lengths of acylated modification[42,43], octanoyl modification is optimal for its activation and biological function[16,17]. The octanoyl group in our structure shows a well-defined density through the cryo-EM map and is observed to bind to hydrophobic cavity II, which comprises residues I178$^{4.60}$, L181$^{4.63}$, L210$^{5.36}$, M213$^{5.39}$, V214$^{5.40}$, and F286$^{6.58}$ from TM4, TM5, and TM6 (Fig. 3d). As seen from the cut view of cavity II,

the octanoyl group definitely fits well in the pocket rather than the other type of fatty acid modification (Supplementary Fig. 8g, h).

The importance of these hydrophobic contacts was validated by single-mutation and cell-based function assays, and our results showed that mutations of I178$^{4.60}$A, L210$^{5.36}$A, and F286$^{6.58}$A decreased the receptor activation induced by ghrelin (Fig. 3d). This finding indicates the critical role of the octanoylation of

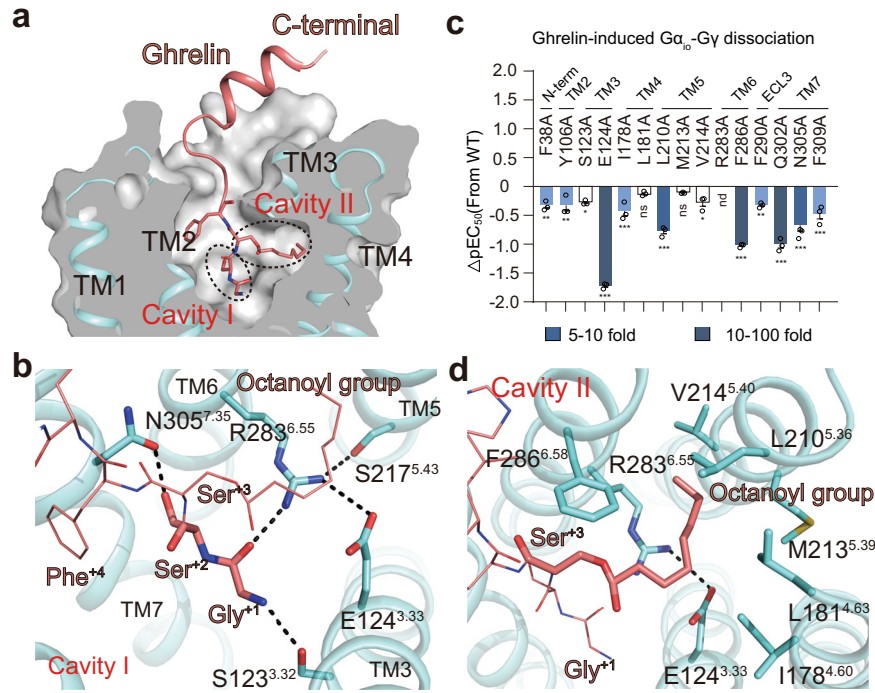

**Fig. 3 Recognition of ghrelin by ghrelin receptor. a** The cutaway surface of the ghrelin receptor bound to the ghrelin structure reveals that ghrelin occupies the bifurcated ligand-binding pocket with the residues $Gly^{+1}$ and $Ser^{+2}$ filling up cavity I and its octanoyl group of $Ser^{+3}$ inserting into cavity II. **b** Detailed polar interaction of both residues $Gly^{+1}$ and $Ser^{+2}$ (aquamarine sticks) with the key residues from cavity I. **c** Ghrelin-induced Gαo1-Gγ dissociation assay. Bars represent differences in calculated potency ($pEC_{50}$) of ghrelin for mutations relative to the WT ghrelin receptor. Data are colored according to the extent of the effect. $*P < 0.1$, $**P < 0.01$ and $***P < 0.001$. ns no significant difference, nd not detected. All data were analyzed by one-way analysis of variance (ANOVA) followed by Dunnett's multiple comparison test, compared with the response of WT ($P = 0.002$, $P = 0.002$, $P = 0.015$, $P < 0.001$, $P < 0.001$, $P = 0.732$, $P < 0.001$, $P = 0.843$, $P = 0.012$, $P < 0.001$, $P = 0.003$, $P < 0.001$, $P < 0.001$ and $P < 0.001$ from left to right). Data represent the mean ± SEM from $n = 3$ biologically independent experiments performed in triplicate. See also Supplementary Figs. 6 and 7. **d** Detailed interaction of the octanoyl group of $Ser^{+3}$ with the key residues from cavity II and salt bridges between $E124^{3.33}$ and $R283^{6.55}$ are clearly observed in the ghrelin-bound structure.

ghrelin in receptor activation as well as the downstream signaling cascade.

Moreover, the contacts between the receptor and ghrelin peptide at the extracellular portion of TM bundles, accompanied by extensive interaction of ECL regions with the C-terminal helix of ghrelin, further stabilize the binding of the receptor with ghrelin (Supplementary Fig. 9a, b). The $F38^{N-term}$, $Y106^{2.67}$, $F290^{ECL3}$, $Q302^{7.32}$, and $F309^{7.39}$ form hydrophobic and polar interactions with the residues $Phe^{+4}$ and $Leu^{+5}$ of ghrelin. Mutation of those residues, especially $Q302^{7.32}$ and $F309^{7.39}$ impaired ghrelin-induced G-protein signaling (Fig. 3d).

**Mechanism of ghrelin-induced activation.** The comparison of the ghrelin-bound complex structure with the neutral antagonist-bound structure shows that the TM bundles at the extracellular portion are similar, except that the helix end of TM7 is tilted outward by ~5.5 Å (Fig. 4a). Moreover, the conformation of the intracellular end of TM bundles is widely changed. The ends of TM3, TM5, TM6, and TM7 shifted by ~2.8, 2.3, 13.4, and 2.9 Å (referring to the Cα of each TM terminal residue: $C146^{3.55}$, $L239^{5.65}$, $L253^{6.25}$, and $L322^{7.52}$), respectively (Fig. 4b). The obvious rearrangement of TM6 as well as the movement of other ends of TMs, which was likewise observed in the Gq-coupled ghrelin receptor structure, make a suitable binding cavity available for the G protein (Fig. 4b and Supplementary Fig. 9c).

The N-terminal part in ghrelin stretches on top of an aromatic cascade formed by $W276^{6.48}$, $F279^{6.51}$, and $F312^{7.42}$, which is referred to as the WFF cluster hereafter. In the neutral antagonist-bound structure, $F312^{7.42}$ was positioned closely and tightly

packed against $F279^{6.51}$ and $W276^{6.48}$. In the ghrelin-bound structure, the side chains of $F312^{7.42}$, $W276^{6.48}$, and $F279^{6.51}$ display significant rotation and displacement toward the extracellular end (Fig. 4c).

$R283^{6.55}$ is situated above $F279^{6.51}$ and connects the WFF cluster with the polar network associated with the binding of ghrelin. Compared with the neutral antagonist-bound structure, the side chain of $R283^{6.55}$ has an obvious swing toward TM5 in the ghrelin-bound structure and appears to trigger local rearrangement of the TM6 bundle. Residues $F279^{6.51}$ and $W276^{6.48}$ shift significantly along with the outward movement of TM6. Notably, $F279^{6.51}$ interacts directly with ghrelin and involves cation-π interactions with $R283^{6.55}$, which also contributes to the movement of $F279^{6.51}$. The swing of $W276^{6.48}$ subsequently switches $F272^{6.44}$ (Fig. 4d), resulting in a cascade of relocations of highly conserved activation motifs such as the $P^{5.50}$-I (V)$^{3.40}$-$F^{6.44}$ motif, D (E)$^{3.49}$-$R^{3.50}$-$Y^{3.51}$ motif, and $N^{7.49}$-$P^{7.50}$-xx-$Y^{7.53}$ motif (Supplementary Fig. 9d–f), leading to the change in TM6 on the intracellular side and enabling the receptor to bind to the G protein. Mutagenesis and functional analyses further verify our hypothesis. Site-directed mutants $E124^{3.33}A$, $R283^{6.55}A$, $F279^{6.51}A$, or $W276^{6.48}A$ almost abolished the activity of the ghrelin receptor, while $F312^{7.42}A$ impaired ghrelin-induced G-protein signaling, implying the critical role of the polar network and the WFF cluster in activation (Fig. 4e).

Aromatic amino acids also contribute to the activation of several other peptide receptors. For instance, $W321^{6.48}$, $F358^{7.42}$, and $Y324^{6.51}$ in NTSR1, which belongs to the ghrelin receptor family (PDB ID: 4XEE), display a similar significance in

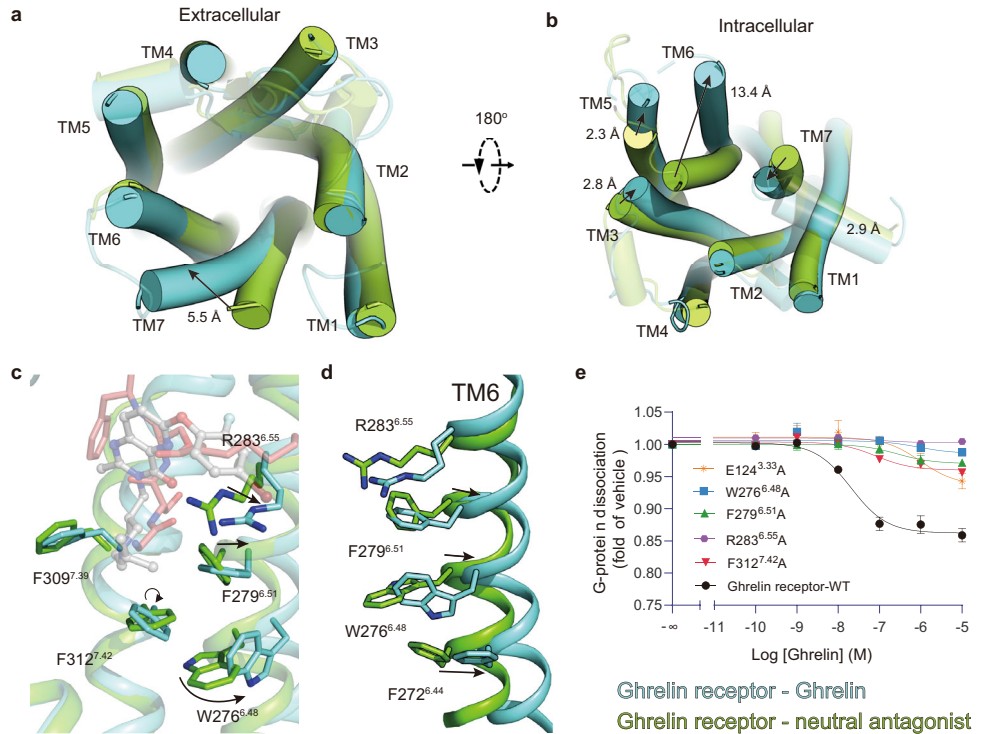

**Fig. 4 Mechanism of ghrelin action on ghrelin receptor. a, b** Structural comparison of agonist-bound with neutral antagonist-bound structures (PDB: 6KO5) reveals remarkable displacements in the extracellular side (**a**) and intracellular side (**b**), resulting in contracting of the orthosteric binding pocket in the extracellular side as well as making a cavity to couple G protein in the intracellular side. **c** Superposition of ghrelin-bound (aquamarine cartoon) and neutral antagonist-bound (forest cartoon) structures shows that the WFF cluster exhibits obvious conformational displacements upon ghrelin binding. **d** The rearrangement of local residues in the backbone of TM6 between ghrelin-bound (aquamarine cartoon) and neutral antagonist-bound (forest cartoon) structures (PDB: 6KO5) shows the cascade changes of residues R283[6.55], F272[6.44], and the WFF cluster. **e** Mutations of W276[6.48], F279[6.51], and F312[7.42] in the ghrelin receptor decreased the activation potency induced by ghrelin. Data represent the mean ± SEM from $n = 3$ biologically independent experiments performed in triplicate. See also Supplementary Figs. 6 and 7.

activation[44]. W86[2.60], Y108[3.32] and Y251[6.51] of CCR5[45] (PDB ID: 7F1R) and W[6.48]-F[6.51]-G[7.42] of 5-HT2aR/2cR[46,47] (PDB ID: 6WH4, 6BQH) were also involved in constitutive activation. The aromatic cascade is also intimately connected with activation in lipid receptors, such as the S1P$_1$ receptor[48] and the CB$_1$ receptor[49]. Collectively, our work demonstrates a possible activation mechanism associated with aromatic clusters for GPCRs.

**Mechanism of inverse agonist action on the ghrelin receptor.** The comparison of the inverse agonist-bound ghrelin receptor structure with agonist-bound and neutral antagonist-bound structures enabled us to visualize the plasticity of the TM domain, reflecting different conformational states of the ghrelin receptor (Supplementary Fig. 5). As observed, the inverse agonist PF-05190457 appears to stabilize the receptor in an inactive conformation with respect to the binding of the G protein (Supplementary Fig. 5b, e). PF-05190457 has distinct chemical moieties with neutral antagonist compound 21, displaying extensive contacts with orthosteric sites. Arm-2 (2-methylimidazo moiety) of PF-05190457 penetrates deeply into the helical bundle and is sandwiched by the WFF cluster (Fig. 5a). As the basal activity of the ghrelin receptor is blocked significantly by inverse agonists, the WFF cluster from the hydrophobic pocket exhibits significant displacement relative to that in agonist-bound and antagonist-bound receptor structures (Fig. 5b). Notably, the side chain of W276[6.48] rotates nearly 180° and tilts ~3 Å toward the extracellular end, and together with the side chain of F279[6.51], it also displays upward movement, therefore pushing out the

extracellular end of TM6 and enlarging the ligand-binding pocket, subsequently impeding the formation of a salt bridge by dragging R283[6.55] away from E124[3.33] (Fig. 5c). These conformational changes also impair the rearrangement of the PIF motif, ERY motif, and NPxxY motif (Supplementary Fig. 9d–f), which is distinct from the conformational changes in agonist-bound and antagonist-bound structures.

To investigate the molecular basis for the aromatic cluster in binding with inverse agonist, we performed cell-based function assays and designed a series of mutations of W276[6.48]A, F279[6.51]A, and F312[7.42]A as well as R283[6.55]A (Figs. 4c and 5e). All four single mutations showed not only decreased receptor activation but also impaired inverse agonism, which is consistent with previous reports that the mutation of aromatic residues from TM6 and TM7 impaired the constitutive activity of the ghrelin receptor[31].

In addition to the change in the hydrophobic portion in the orthosteric site, the rearrangement of the polar network occurs as inverse agonist binding. Through structural comparison with ghrelin-bound, the PF-05190457-bound complex structure shows breaking of the salt bridge between R283[6.55] and E124[3.33], breaking of the weak polar interaction between R102[2.63] and Q120[3.29], and newly established polar interactions between the inverse agonist and residues S308[7.38] and D99[2.60] (Fig. 5d). Notably, E124[3.33]A and R283[6.55]A mutants impaired the inverse agonism potency of PF-05190457 (Fig. 5e and Supplementary Fig. 7a), suggesting that both E124[3.33] and R283[6.55] (referred to as the E-R motif) should play an important role in the constitutive activity of the ghrelin receptor and the action of the inverse agonist.

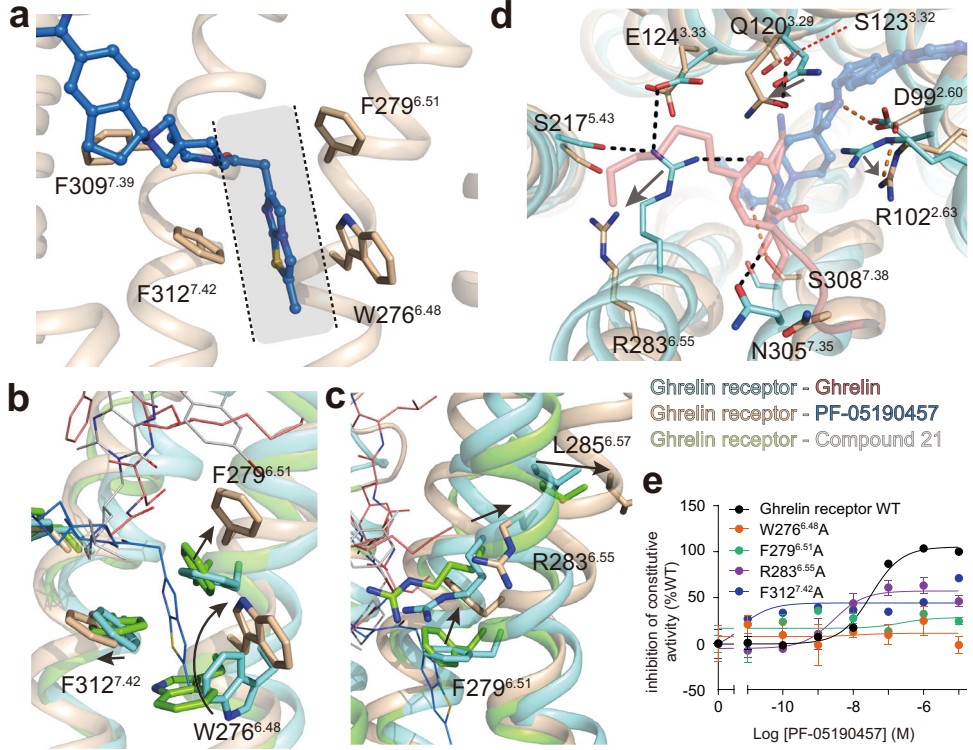

**Fig. 5 Detailed mechanism of inverse agonism action on the ghrelin receptor. a** The arm-2 group of PF-05190457 was observed to insert into the core of the receptor and was sandwiched by side chains from the hydrophobic cluster W276$^{6.48}$-F279$^{6.51}$-F312$^{7.42}$ (WFF cluster). **b** Superposition of PF-05190457-bound (wheat cartoon) and ghrelin-bound (aquamarine cartoon) structures shows that the WFF cluster exhibits obvious conformational displacements upon inverse agonist binding. **c, d** The inverse agonist PF-05190457 pushes outward movements of the extracellular end of TM6 (**c**), destroying the key polar interactions involved in activation (**d**). The polar networks are shown as black dotted lines (ghrelin-bound) and orange dotted lines (PF-05190457-bound). **e** Mutations of W276$^{6.48}$, F279$^{6.51}$, and F312$^{7.42}$ in the ghrelin receptor significantly reduced the potency of inverse agonism. Data represent the mean ± SEM from $n = 3$ biologically independent experiments performed in triplicate. See also Supplementary Fig. 6.

## Discussion

The ghrelin receptor is reported to have high constitutive activity[25]. To better understand the constitutive activity mechanism of the ghrelin receptor, we compared ghrelin receptors with different states as well as with previously described receptors with high basal activation, such as GPR52. The structure of GPR52 contains a built-in ECL2 region that is inserted into the orthosteric binding pocket[50] (Supplementary Fig. 10a). In particular, Y185$^{ECL2}$ and H186$^{ECL2}$ act as built-in "agonist" for activating GPR52, thus resulting in a high level of basal activity of the receptor[50] (Supplementary Fig. 10b). The structural superposition of GPR52 with the ghrelin receptor indicates that the ECL2 region of ghrelin receptor sits over the ligand-binding pocket (Supplementary Fig. 10c). Together with the structural comparison with ghrelin receptors with different types of ligands, our results suggest that polar networks such as the E-R motif and another notable hydrophobic WFF cluster should be required for the constitutive activity of receptors. More interestingly, our careful inspection of sequence alignment revealed that the equivalent E-R motif and aromatic cluster were also found in NTSR1, NTSR2, and GPR39 (Supplementary Fig. 11a). Previous reporting showed that NTSR2 and GPR39 display high constitutive activity[31]. Consistently, the ghrelin receptor has ~50% maximal constitutive activity[10], and the basal activity of NTSR2 was observed to be similar to that of the ghrelin receptor[31], yet the constitutive activity was enhanced when mutating the F7.42A in NTSR1. To further figure out the functional roles of the equivalent E-R motif and WFF cluster in NTSR1, we generated several mutations and carried out constitutive activity assays. The results of our measurement reveal that residues at positions 3.33

and 6.51 in NTSR1 should be engaged in constitutive activity of the receptor (Supplementary Fig. 11b)[51]. Therefore, we speculated that the receptors containing E-R motif and WFF cluster may contribute to their basal activity.

Numerous GPCRs have been reported to contain constitutive activity, and several structures of inverse agonist-bound GPCRs have been determined so far[35–37,47]. As shown in Supplementary Fig. 12b, inverse agonists in DRD2, 5-HT2aR, 5-HT2cR, and ghrelin receptor appear to extend deeply into the orthosteric pocket, forming directly hydrophobic interactions with the indole ring of the "switch" residue W$^{6.48}$. We note that the aromatic moiety of inverse agonists packs against either the left or right of W$^{6.48}$ in the receptors mentioned above (Supplementary Fig. 12b). Methiothepin, RIT, and risperidone were observed to bind with the hydrophobic cavity on the left of W$^{6.48}$ in 5-HT2aR, 5-HT2cR, and DRD2, forming direct interactions with the W$^{6.48}$-F$^{6.51/6.52}$-F$^{5.47}$ clusters. Whereas the PF-05190457 and IRL2500 bind to another hydrophobic cavity on the right of W$^{6.48}$ in the ghrelin receptor and ETB receptor. Compared with the activated 5-HT2cR and DRD2 structures, the hydrophobic W$^{6.48}$-F$^{6.51/6.52}$-F$^{5.47}$ cluster also notably displaced the structures of inverse agonist-bound receptors (Supplementary Fig. 12d–f). The deep binding conformation of the inverse agonist impeded the side-chain rotations of W$^{6.48}$ and F$^{6.44}$, restricting the conformational change of TM6 from inactive to active state.

In addition, sequence alignment and structural comparison of ghrelin receptors with 5-HT2aR, 5-HT2cR, DRD2 and ETB receptors revealed that the E$^{3.33}$-R$^{6.55}$ motif is not conserved (Supplementary Fig. 12a, c), and the distance between TM3 and TM6 appears to play a role in ligand selectivity. Collectively, in

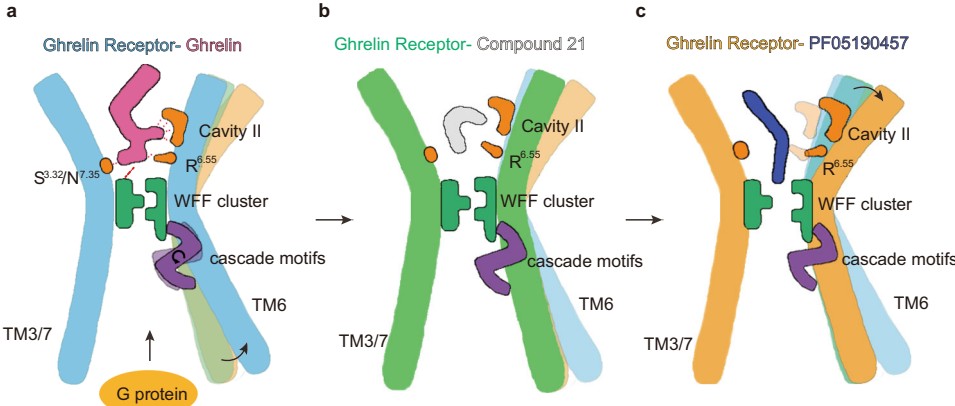

**Fig. 6 Summary model of ghrelin receptor bound with inverse agonist, agonist, and antagonist.** Simple model for PF-05190457 (**a**), ghrelin (**b**), and compound 21 (**c**) bound to the ghrelin receptor. Some of the essential residues and modules are marked with different colors and patterns. The cascade motifs referred to the combination of the P–I–F motif, E-R-Y motif, and NPxxY motif.

the ghrelin receptor family (including ghrelin receptor, NTSR1, NTSR2, and GPR39), the conserved $E^{3.33}$-$R^{6.55}$ motif together with the WFF cluster is suggested to be essential for inverse agonist binding as well as agonist activation.

When our manuscript was under review, another study about the Gq-coupled ghrelin receptor was published, and it reported the structure of the ghrelin-bound receptor complex[39]. Obviously, $Gly^{+1}$ and $Ser^{+2}$ of ghrelin inserted into the same orthosteric site of the receptor, whether binding to either Go or Gq. The backbone of the ghrelin peptide resembled the identical conformation. However, a subtle difference was the position of the octanoyl groups at $Ser^{+3}$ in the two ghrelin peptides (Supplementary Fig. 8b). The acyl chain of $Ser^{+3}$ extends towards TM5 in the Go-bound receptor, while the octanoic tail stretches horizontally toward the gap between TM4 and TM5 and five carbons of the octanoyl group can be placed in the density in the Gq-bound state, reflecting a presumptive dynamic character of acyl modification in the ghrelin receptor (Supplementary Fig. 8b). Coincidentally, previous literature has proven the dynamic characteristics of ghrelin receptors using NMR spectroscopy[52]. Compared with the available structure of the ghrelin peptide, the disordered C-terminus of the peptide folds into α-helix when ghrelin binds to its receptor (Supplementary Fig. 8c), displaying the dynamic conformation of the peptide. Therefore, it is not hard to speculate that the octanoyl group of ghrelin could behave in variable conformations. In agreement with this speculation, molecular simulation courses and NMR data indicated some degree of conformational and local dynamics of the ghrelin peptide in the ligand-binding pocket[53].

In this study, we present the structures of ghrelin receptors in complex with the inverse agonist PF-05190457 as well as in complex with endogenous ghrelin and Go protein coupling. Combined with the inverse agonist-, agonist- and antagonist-bound structures, the molecular mechanism of the activation of ghrelin receptor has gradually emerged, and it can be summarized as follow: The binding of ghrelin peptide to the receptor contracts the extracellular TM bundles and may change the orientation of the polar network and hydrophobic core including E-R motif and WFF cluster, further expanding the intracellular end of TM6 and allowing the ghrelin receptor to interact with G proteins by rearranging the cascade motifs (including PIF motif, ERY motif, and NPxxY motif) (Fig. 6a). In addition, the binding of the antagonist occupies the orthosteric site and maintains the conformation of the E-R motif and WFF cluster (Fig. 6b). However, the binding of PF-05190457 will break the tight conformation of the WFF cluster, push against the extracellular TM6 bundle, and

contract the intracellular end of the TM6 helix to close the G-protein-binding pocket (Fig. 6c).

Taken together, the discovery of constitutive receptor activity and inverse agonism extends the knowledge of GPCR pharmacology, which may help pharmacologists to achieve a greater degree of control over receptor function. Our study provides insights into the detailed mechanism of ghrelin receptor binding to inverse agonists and agonists, and it may pave the way for designing specific ligands targeting ghrelin receptors.

## Methods

**Construct design and expression**. For crystallization construct ghrelin receptor-bRIL, the wild-type human ghrelin receptor cDNA gene (UniProt accession: Q92847) was cloned into a modified pFastBac1 (Invitrogen) baculovirus expression vector with the haemagglutinin (HA) signal sequence followed by a Flag epitope tag at the N-terminus and a 10× His tag at the C-terminus. To facilitate receptor expression and crystallization, the N-terminal 34 residues and C-terminal residues after P342 were removed, TEV protease recognition sites were introduced before the residue Leu35 at N-terminus and before the residue F343 at C-terminus. Residues $R244^{5.70}$-$S252^{6.24}$ of the intracellular loop 3 (ICL3) were replaced with the thermostabilized apocytochrome $b_{562}$RIL (bRIL) (PDB: 1M6T). Furthermore, mutations $T130^{3.39}$K and $N188^{ECL2}$Q were introduced to improve thermostability and homogeneity. The final construct of the ghrelin receptor was transfected into DH10Bac™ *Escherichia coli* to produce a recombinant baculovirus with the Bac-to-Bac system (Invitrogen). The recombinant baculovirus was used to infect Sf9 insect cell culture at a cell density of $2.5 × 10^6$ cells per ml⁻¹. Infected cells were grown for 48 h at 27 °C before harvesting, and the cell pellets were stored at −80 °C for future use.

For cryo-EM constructs, the full-length of wild-type human ghrelin receptor was subcloned into pFastBac1 vector with an N-terminal FLAG tag and C-terminal 10×His tag. We used the NanoBiT tethering strategy, in which the C-terminus of the ghrelin receptor was directly attached to the LgBiT subunit followed by a TEV protease cleavage site and a double MBP tag. Gβ1 was fused with a C-terminal HiBiT, together with Gγ2 were cloned into pFastBac dual vector. An engineered human Gαo1 with Gαo1 H domain deletion, named miniGαo1 was cloned into pFastBac1 according to published literature[33]. The ghrelin-induced ghrelin receptor activity was measured by NanoBiT-G-protein dissociation assay using a chimeric Gαio protein according to the previous publication except for replacement Gαqo with Gαio[54]. The Gαio were generated by replacing the six amino acids of the C-terminal of Gαi with those from GαoA1. Other constructs including the full-length and various site-directed mutagenesis human ghrelin receptors were cloned into pcDNA3.1 vector for NanoBiT-G-protein dissociation assay. Ghrelin receptor, miniGαo1, and Gβ1γ2 were co-expressed in Sf9 insect cells (Expression System) using the Bac-to-Bac baculovirus expression system (ThermoFisher). The cell pellets were collected by centrifugation 48 h post infection and stored at −80 °C until use.

**Purification of ghrelin receptor-bRIL protein**. The cell pellets were lysed using a Dounce homogenizer in a hypotonic buffer containing 10 mM HEPES (pH 7.5), 20 mM KCl, 10 mM MgCl₂, 160 μg/ml benzamidine, and 100 μg/ml leupeptin. The cell membranes were isolated by ultracentrifugation at 100,000×g for 30 min at 4 °C. Washing of the membranes was performed by two rounds of Dounce homogenization and centrifugation in a high-osmolarity buffer containing 10 mM

HEPES (pH 7.5), 1.0 M NaCl, 20 mM KCl, and 10 mM $MgCl_2$, 160 μg/ml benzamidine, and 100 μg/ml leupeptin. Purified membranes were incubated with 10 μM PF-05190457 (Tocris) and 2 mg/ml iodoacetamide in hypotonic buffer for 2 h. The membranes were then solubilized in a solubilization buffer containing 10 μM PF-05190457, 50 mM HEPES (pH 7.5), 500 mM NaCl, 10%(v/v) glycerol, 1%(w/v) n-dodecyl-ß-D-maltopyranoside (DDM), 0.2% cholesteryl hemisuccinate (CHS), and 0.2% (w/v) sodium cholate,160 μg/ml benzamidine, 100 μg/ml leupeptin for 2.5 h at 4 °C, followed by ultracentrifugation at 125,000 × $g$ for 30 min at 4 °C. The supernatant was incubated with TALON IMAC resin (TaKaRa) in a batch overnight at 4 °C. After binding, the resin was collected by centrifugation at 500 × $g$ for 5 min, resuspended with wash buffer (50 mM HEPES pH 7.5, 500 mM NaCl, 0.1% lauryl maltose neopentyl glycol (LMNG; Anatrace). 0.02% sodium cholate, 0.02% CHS, 5% (v/v) glycerol) supplemented with 160 μg/ml benzamidine, 100 μg/ml leupeptin, 10 mM imidazole, 1 μM PF-05190457 and was repeated one more time. Then the resin was loaded on a glass column and slowly washed with wash buffer containing 160 μg/ml benzamidine, 100 μg/ml leupeptin, 15 mM imidazole, and 1 μM PF-05190457 for gradually exchanging to LMNG. The protein was eluted with 10 ml wash buffer containing 200 mM imidazole, 10 μM PF-05190457. The N-terminal FLAG tag and C-terminal 10× His tag were cleaved by TEV protease for 6 h at 4 °C. Finally, the receptor was run on a Superdex 200 size-exclusion column (GE Healthcare) with buffer containing 20 mM HEPES (pH 7.5), 150 mM NaCl, 0.02% LMNG, 0.004% CHS, and 10 μM PF-05190457.

**Crystallization, data collection, and structure determination**. The purified receptor—PF-05190457 complex was concentrated to >60 mg/ml using a 100-kDa cutoff Vivaspin concentrator (Sartorius). The initial crystallization screen was set up using the lipidic cubic phase (LCP) method[55]. The sample of the complex was mixed with the lipid (monoolein and cholesterol at 10:1 by mass) at a weight ratio of 2:3 using a syringe mixing apparatus at room temperature[56]. The mesophase was then dispensed into glass sandwich plates in 40 nl drops and overlaid with 800 nl precipitant solution using a Gryphon LCP robot (Art Robbins Instruments). The full-size crystals of ghrelin receptor were grown over 1 week at 20 °C in the following overlay precipitant condition: 25–6% PEG300, 100 mM HEPES pH 7.0, 100–150 mM $NH_4F$. Finally, the LCP crystals were frozen quickly in liquid nitrogen after harvesting.

The X-ray diffraction data were collected at beamline 32XU at SPring-8, Hyogo, Japan, using a beam size of 10 μm and a Pilatus3 6 M detector (X-ray wavelength 1.0000 Å). The data-collection strategy was designed and performed on the basis of initial raster results as described previously[57]. Full datasets of PF-05190457-bound ghrelin receptor were assembled from 23 crystals owing to radiation damage of crystals. Diffraction images were indexed, integrated, and scaled using XDS[58] and merged using SCALA. The resolution limit was set to 2.94 Å. Data-collection statistics are shown in Supplementary Table 1. The structure of the PF-05190457-bound ghrelin receptor was determined by molecular replacement with PHASER[59], using ghrelin receptor (receptor only, PDB: 6KO5) and bRIL (PDB: 1M6T) as independent search models. The refinement and manual building were performed by PHENIX[59] and COOT[60], respectively. The refinement parameter for ligand PF-05190457 was generated with the PRODRG[61] web server. Refinement statistics are reported in Supplementary Table 1. Figures were generated using PyMOL (https://pymol.org).

**Ghrelin-bound ghrelin receptor-Go complex formation and purification**. The cell pellets were lysed in a buffer containing 20 mM HEPES, pH 7.5, 100 mM NaCl, and 2 mM $MgCl_2$ supplemented with EDTA-free protease inhibitor cocktail (Bimake) by dounce homogenization. The complex formation was initiated by addition of 10 μg/mL scFv16, 50 mU/mL apyrase (NEB), and 100 mM Ghrelin. After incubation at room temperature for 1.5 h, the membranes were solubilized by addition of 0.5% (w/v) lauryl maltose neopentyl glycol (LMNG, Anatrace) and 0.1% (w/v) cholesteryl hemisuccinate TRIS salt (CHS, Anatrace) for 2 h at 4 °C. The supernatant was isolated by centrifugation at 30,000 × $g$ for 30 min and then incubated 1 h at 4 °C with pre-equilibrated MBP resin. After binding, the resin was washed with 15 column volumes of 20 mM HEPES pH 7.5, 100 mM NaCl, 2 mM $MgCl_2$, 0.01% (w/v) LMNG, 0.002% (w/v) CHS, and 10 mM Ghrelin. The complex was eluted with 5 column volumes of 20 mM HEPES pH 7.5, 100 mM NaCl, 2 mM $MgCl_2$, 0.01% (w/v) LMNG, 0.002% (w/v) CHS, 10 mM maltose, and 10 mM ghrelin.

The protein was then concentrated and loaded onto a Superose™ 6 Increase column (GE Healthcare) pre-equilibrated with buffer containing 20 mM HEPES pH 7.5, 100 mM NaCl, 0.00075% (w/v) LMNG, 0.00025% (w/v) glyco-diosgenin (GDN, Anatrace), 0.0002% (w/v) CHS and 10 mM ghrelin. The fractions for the monomeric complex were collected and concentrated for electron microscopy experiments.

**Cryo-EM grid preparation and data collection**. For the cryo-EM grids preparation, 3 uL purified protein ghrelin receptor-miniGαo1-Gβ1γ2 complex at the concentration of about 15 mg/mL were applied individually to a glow discharged holey carbon EM grid (Quantifoil, Au300 R1.2/1.3) in a Vitrobot chamber (FEI Vitrobot Mark IV). The grids were blotted for 3.0 s with a blot force of 3 at 4 °C,

100% humidity, and then plunge-frozen in liquid ethane using Vitrobot Mark IV (Thermo Fischer Scientific). Cryo-EM data collection was performed on a Titan Krios at 300 kV accelerating voltage in the Center of Cryo-Electron Microscopy, Zhejiang University (Hangzhou, China). Micrographs were recorded using a Gatan K2 Summit detector in counting mode with a pixel size of 1.014 Å using the SerialEM software. The total exposure time was 8 s and 40 frames were recorded per micrograph.

**Cryo-EM data processing**. Image stacks for ghrelin receptor-miniGαo1-Gβ1γ2 complex were subjected to beam-induced motion correction using MotionCor2.1[62]. Contrast transfer function (CTF) parameters were estimated by Gctfv1.18[63]. The following data processing was performed using RELION-3.0-beta2[64]. Auto-picking automated particle selection using Gaussian blob detection produced 2,824,307 particles. The particles were subjected to four rounds of 3D classifications on the complex and selected the best-resolved class. Further 3D classification focusing the alignment on the ghrelin receptor produced a high-quality subset accounting for 230,306 particles, which were subsequently subjected to 3D refinement, CTF refinement, and Bayesian polishing. The final refinement generated a map with an indicated global resolution of 2.8 Å at a Fourier shell correlation of 0.143. Local resolution was determined using the Bsoft package with half maps as input maps[65].

**Cryo-EM model building, refinement, and validation**. The crystal structure of the ghrelin receptor[30] (PDB: 6KO5) was used as an initial model for model rebuilding and refinement against the electron microscopy map of ghrelin receptor-miniGαo1-Gβ1γ2 complex. The BCM–GPR97–Go complex[33] (PDB 7D76) was to generate the initial models of Go, Gβγ, and scFV16. Ligand and lipids coordinate and geometry restraints were generated using phenix. elbow. Models were docked into the EM density map using UCSF Chimera (https://www.cgl.ucsf.edu/chimera/). This starting model was then subjected to iterative rounds of manual adjustment and automated refinement in Coot[66] and Phenix10[59], respectively.

The final refinement statistics were validated using the module "comprehensive validation (cryo-EM)" in PHENIX. To monitor the potential over-fitting in model building, FSCwork and FSCtest were determined by refining the 'shaken' models against unfiltered half-map-1 and calculating the FSC of the refined models against unfiltered half-map-1 and half-map-2. Structural figures were prepared in Chimera (https://www.cgl.ucsf.edu/chimera), ChimeraX11[67,68] and PyMOL (https://pymol.org/). The final refinement statistics are provided in Supplementary Table 2.

**IP1 accumulation assays**. To measure the activity of inverse agonist PF-05190457 (Tocris, Cat. No. 6350), we carried out IP1 accumulation assays for ghrelin receptor independent of agonist. The agonist ghrelin and antagonist compound 21 were also performed as the control. The cDNA of ghrelin receptor subcloned into pcDNA3.1(+) expression vector with a HA signal sequence followed by a Flag tag at the N-terminus. Point mutations used in our study were generated by using Q5 site-Directed Mutagenesis kit (NEB). All the constructs were verified by sequencing. The constructs were expressed in HEK293 cells using PEI transfection reagent (YEASEN, 40815es03) according to the manufacturer's instruction. Cells were harvested 48 h post transfection.

IP accumulation was measured using Cisbio IP1 assay kit (Cisbio, 62IPAPEB), which has been previously described[32]. In brief, the harvested cells were distributed in a low volume 96-well dish and stimulated with increased concentration of inverse agonist PF-05190457, agonist ghrelin peptide or antagonist compound 21 containing 10 nM ghrelin peptide at 37 °C for 2 h. After that, d2-labeled IP1 and cryptate-labeled anti-IP1 monoclonal antibody dissolved in Lysis Buffer were added to the wells. After 1 h incubation at room temperature. IP1 was quantified using Synergy H1 microplate reader (BioTek) with excitation at 320 nm, emission at 620 nm, and 665 nm. The accumulation of IP1 was calculated according to a standard dose–response curve in GraphPad Prism 8 (GraphPad Software). Data were represented as the mean ± SEM from three independent experiments and all experiments were repeated at least three times.

**NanoBiT-G-protein dissociation assay**. G-protein activation was detected by a NanoBiT-G-protein dissociation assay[69]. In brief, HEK293T cells were plated in each well of a six-well plate at a concentration of 0.3 million/well (2 mL per well). Plasmid transfection was performed with a mixture of 400 ng Gαio-LgBiT, 500 ng Gγ-SmBiT, 500 ng Gβ, 600 ng ghrelin receptor by Lipofectamine 2000 (Thermo-Fisher Scientific) in 200 μL of Opti-MEM (Gibco). After 1 day of transfection, cells in the six-well plate were digested and resuspended in complete medium DMEM (5% FBS, 1% antibiotic) and plated in a 96-well flat-bottomed white microplate. After 24 h, the cells were washed twice with D-PBS and incubated in 40 μL of 5 μM coelenterazine 400a (Maokangbio) solution diluted 0.01% BSA and 5 mM HEPES (pH 7.5)-containing HBSS (assay buffer) for 1 h at room temperature. Baseline luminescence was measured using a luminescent microplate reader (Tecan). Ghrelin (5×, diluted in the assay buffer) was added to the cells (10 μL) and incubated for 3–5 min at room temperature before the second measurement. The ligand-induced signal ratio was normalized to the baseline luminescence and fold-

change signals over vehicle treatment were used to show G-protein dissociation response.

**Enzyme-linked immunosorbent assay (ELISA)**. Cell surface expression of the receptor subunits was detected by ELISA. Plasmids corresponding to WT and mutant ghrelin receptors were transfected as described above. After transfection, cells were re-seeded onto cell adherent reagent (Applygen) treated 96-well plates at a density of $3 \times 10^4$ cells per well. Twenty-four hours later, cells were washed with PBS and fixed with 10% formaldehyde for 10 min followed by three times washing with PBS. Following fixation, cells were blocked with blocking buffer (1% BSA in PBS) for 1 h at RT. Afterward, plates were incubated with a 1:10,000 dilution of anti-FLAG M2 HRP-conjugated monoclonal antibody (SigmaAldrich) in blocking buffer for another 1 h at RT. After careful washing, 80 μL/well diluent SuperSignal Elisa Femto Maximum Sensitivity Substrate (ThermoFisher Scientific) was added, and the luminescence was measured using a luminescence microplate reader (Tecan).

**Synthesis**. All solvents and chemicals used for compound **21** synthesis were reagent grade and supplied by commercial sources. Silica gel thin-layer chromatography was performed on pre-coated plates GF-254 (Qingdao HaiYang, China). The compound purity and characterization were established by a combination of liquid chromatography-mass spectroscopy (LCMS) and nuclear magnetic resonance (NMR) analytical techniques. The LCMS using Waters Acquity_Arc-2489-Qda (Column: Xbridge C18; Column size: 3.5 μm 2.1 × 50 mm, mobile phase: A: (0.1% FA) $H_2O$, B: MeCN; gradient B%: as Acq. operating in ES (+) ionization mode; $T = 40\,°C$; flow rate = 1.0 mL/min; detected wavelength, 220 nm). The $^1H$-NMR spectra were recorded on Burker AVANCEIII 600 MHz (Bruker Company). Chemical shifts are provided in ppm (δ). Compound **21** was prepared as follow (see also in Supplementary Fig. 13a):

Briefly, a mixture of 3-amino-6-chloropyridine-2-carboxylicacid (0.5 g, 2.9 mmol), tert-butyl (R)-3-(aminomethyl)piperidine-1-carboxylate (0.8 g, 3.7 mmol), BOP (Tri(dimethylamino)benzotriazol-1-yloxyphosphonium hexafluorophosphate) (1.5 g, 3.4 mmol) and TEA (0.9 g, 8.7 mmol) in DCM (12.5 mL) was stirred overnight at room temperature. The reaction mixture was washed with saturated sodium bicarbonate solution, saturated sodium chloride solution, and concentrated. The residue was purified by thin-layer chromatography (TLC) (DCM/MeOH = 15/1) to give the title compound **3** (1 g, 93.5%) as a yellow solid.

Compound **3** (1 g, 2.7 mmol) was dissolved in Ethyl orthoacetate (2.2 g, 13.6 mmol). Added HOAc (163 mg, 2.7 mmol) in the solution and stirred at 135 °C for 3 h. The reaction mixture was concentrated in a vacuum to give the title compound **4** (0.86 g, 81.1%) as a white solid. Compound **4** (0.86 g, 2.2 mmol), 4-bromo-2-fluorophenol (1.25 g, 6.6 mmol) and Caesium carbonate (2.2 g, 6.6 mmol) were dissolved in DMF (10 mL) and stirred 6 h at 90 °C. The reaction mixture was extracted by the saturated sodium bicarbonate and EAC. The residue was dried over $Na_2SO_4$ and concentrated in a vacuum to afford the title compound **6** (1 g, 83.3%) as yellow solid.

Added EA/HCl (3 mL) to the solution of compound **6** (0.5 g 0.9 mmol) dissolved in EA (3 mL) stirred 0.5 h at room temperature. The reaction mixture was concentrated in a vacuum. The residue was dissolved in $CH_3CN$ (10 mL), then added 2-iodopropane (0.5 g, 2.7 mmol) and $K_2CO_3$ (0.5 g, 4.6 mmol) stirred 6 h at 70 °C. The reaction mixture was extracted with EAC, dried over $Na_2SO_4$ and concentrated. The residue was purified by TLC (DCM/MeOH = 20/1) to afford the compound **21** (0.3 g 67.1%) as a white solid. The compound **21** was measured to have 98.8% purity before the experiment. $^1H$-NMR (600 MHz, CDCl3): δ7.98 (d, J = 13.2 Hz, 1 H), 7.26–7.39 (m, 4 H), 4.03 (s, 2 H), 2.69–2.73 (m, 6 H), 2.13–2.24 (m, 3 H), 1.69–1.74 (m, 3 H), 1.23–1.27 (m, 1 H), 1.01–1.04 (dd, J = 4.8,9 Hz, 6 H). MS $m/z$ (ES$^+$) = 489.13 (Supplementary Fig. 13b–d).

**Statistics and reproducibility**. All functional study data were analyzed using Prism 8 (GraphPad) and presented as means ± SEM. from at least $n = 3$ biologically independent experiments performed in triplicate. Concentration–response curves were evaluated with a standard dose–response curve. Statistical differences were determined by two-sided, one-way ANOVA with Dunnett's multiple comparisons test.

**Reporting summary**. Further information on research design is available in the Nature Research Reporting Summary linked to this article.

## Data availability

The structural data generated in this study have been deposited in the Protein Data Bank (http://www.pdb.org/) under accession number 7F83 for the PF-05190457-ghrelin receptor, 7W2Z and Electron Microscopy Data Bank (EMDB) accession number EMD-32268 for the ghrelin–ghrelin receptor-Go complex. All the other data generated in this study are provided in the Supplementary information and source data files. Source data are provided with this paper.

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

# ARTICLE

31. Holst, B. et al. Common structural basis for constitutive activity of the ghrelin receptor family. *J. Biol. Chem.* **279**, 53806–53817 (2004).

32. Yan, W. et al. Structure of the human gonadotropin-releasing hormone receptor GnRH1R reveals an unusual ligand binding mode. *Nat. Commun.* **11**, 5287 (2020).

33. Ping, Y. Q. et al. Structures of the glucocorticoid-bound adhesion receptor GPR97-G(o) complex. *Nature* **589**, 620–626 (2021).

34. Duan, J. et al. Cryo-EM structure of an activated VIP1 receptor-G protein complex revealed by a NanoBiT tethering strategy. *Nat. Commun.* **11**, 4121 (2020).

35. Wang, S. et al. Structure of the D2 dopamine receptor bound to the atypical antipsychotic drug risperidone. *Nature* **555**, 269–273 (2018).

36. Nagiri, C. et al. Crystal structure of human endothelin ET(B) receptor in complex with peptide inverse agonist IRL2500. *Commun. Biol.* **2**, 236 (2019).

37. Shao, Z. et al. High-resolution crystal structure of the human CB1 cannabinoid receptor. *Nature* **540**, 602–606 (2016).

38. Kung, D. W. et al. Identification of spirocyclic piperidine-azetidine inverse agonists of the ghrelin receptor. *Bioorg. Med. Chem. Lett.* **22**, 4281–4287 (2012).

39. Wang, Y. et al. Molecular recognition of an acyl-peptide hormone and activation of ghrelin receptor. *Nat. Commun.* **12**, 5064 (2021).

40. Ring, A. M. et al. Adrenaline-activated structure of β2-adrenoceptor stabilized by an engineered nanobody. *Nature* **502**, 575–579 (2013).

41. Xiao, P. et al. Ligand recognition and allosteric regulation of DRD1-Gs signaling complexes. *Cell* **184**, 943–956.e918 (2021).

42. Ohgusu, H. et al. Ghrelin O-acyltransferase (GOAT) has a preference for n-hexanoyl-CoA over n-octanoyl-CoA as an acyl donor. *Biochem. Biophys. Res. Commun.* **386**, 153–158 (2009).

43. Gutierrez, J. A. et al. Ghrelin octanoylation mediated by an orphan lipid transferase. *Proc. Natl Acad. Sci. USA* **105**, 6320–6325 (2008).

44. Krumm, B. E., White, J. F., Shah, P. & Grisshammer, R. Structural prerequisites for G-protein activation by the neurotensin receptor. *Nat. Commun.* **6**, 7895 (2015).

45. Zhang, H. et al. Structural basis for chemokine recognition and receptor activation of chemokine receptor CCR5. *Nat. Commun.* **12**, 4151 (2021).

46. Kim, K. et al. Structure of a hallucinogen-activated Gq-coupled 5-HT(2A) serotonin receptor. *Cell* **182**, 1574–1588.e1519 (2020).

47. Peng, Y. et al. 5-HT(2C) receptor structures reveal the structural basis of GPCR polypharmacology. *Cell* **172**, 719–730.e714 (2018).

48. Hanson, M. A. et al. Crystal structure of a lipid G protein-coupled receptor. *Science* **335**, 851–855 (2012).

49. Hua, T. et al. Crystal structures of agonist-bound human cannabinoid receptor CB(1). *Nature* **547**, 468–471 (2017).

50. Lin, X. et al. Structural basis of ligand recognition and self-activation of orphan GPR52. *Nature* **579**, 152–157 (2020).

51. Deluigi, M. et al. Complexes of the neurotensin receptor 1 with small-molecule ligands reveal structural determinants of full, partial, and inverse agonism. *Sci. Adv.* **7**, eabe5504 (2021).

52. Vortmeier, G. et al. Integrating solid-state NMR and computational modeling to investigate the structure and dynamics of membrane-associated ghrelin. *PLoS ONE* **10**, e0122444 (2015).

53. Ferré, G. et al. Structure and dynamics of G protein-coupled receptor-bound ghrelin reveal the critical role of the octanoyl chain. *Proc. Natl Acad. Sci. USA* **116**, 17525–17530 (2019).

54. Inoue, A. et al. Illuminating G-protein-coupling selectivity of GPCRs. *Cell* **177**, 1933–1947.e1925 (2019).

55. Cherezov, V. Lipidic cubic phase technologies for membrane protein structural studies. *Curr. Opin. Struct. Biol.* **21**, 559–566 (2011).

56. Caffrey, M. & Cherezov, V. Crystallizing membrane proteins using lipidic mesophases. *Nat. Protoc.* **4**, 706–731 (2009).

57. Li, X. et al. Crystal structure of the human cannabinoid receptor CB2. *Cell* **176**, 459–467.e413 (2019).

58. Kabsch, W. XDS. *Acta Crystallogr D Biol. Crystallogr* **66**, 125–132 (2010).

59. Adams, P. D. et al. PHENIX: a comprehensive Python-based system for macromolecular structure solution. *Acta Crystallogr D Biol. Crystallogr* **66**, 213–221 (2010).

60. Emsley, P., Lohkamp, B., Scott, W. G. & Cowtan, K. Features and development of Coot. *Acta Crystallogr D Biol. Crystallogr* **66**, 486–501 (2010).

61. Schüttelkopf, A. W. & van Aalten, D. M. PRODRG: a tool for high-throughput crystallography of protein-ligand complexes. *Acta Crystallogr D Biol. Crystallogr* **60**, 1355–1363 (2004).

62. Zheng, S. Q. et al. MotionCor2: anisotropic correction of beam-induced motion for improved cryo-electron microscopy. *Nat. Methods* **14**, 331–332 (2017).

63. Zhang, K. Gctf: real-time CTF determination and correction. *J. Struct. Biol.* **193**, 1–12 (2016).

64. Scheres, S. H. RELION: implementation of a Bayesian approach to cryo-EM structure determination. *J. Struct. Biol.* **180**, 519–530 (2012).

65. Heymann, J. B. Guidelines for using Bsoft for high resolution reconstruction and validation of biomolecular structures from electron micrographs. *Protein Sci.* **27**, 159–171 (2018).

66. Emsley, P. & Cowtan, K. Coot: model-building tools for molecular graphics. *Acta Crystallogr D Biol. Crystallogr* **60**, 2126–2132 (2004).

67. Goddard, T. D. et al. UCSF ChimeraX: meeting modern challenges in visualization and analysis. *Protein Sci.* **27**, 14–25 (2018).

68. Pettersen, E. F. et al. UCSF ChimeraX: structure visualization for researchers, educators, and developers. *Protein Sci.* **30**, 70–82 (2021).

69. Kato, H. E. et al. Conformational transitions of a neurotensin receptor 1-G(i1) complex. *Nature* **572**, 80–85 (2019).

## Acknowledgements

This work was supported by Natural Science Foundation of China grant 31972916 to Z.S.; Ministry of Science and Technology of China grant 2019YFA0508800 (Z.S. and Y.Z.); Science and Technology department of Sichuan Province 2020YJ0208 (Z.S.); National Natural Science Foundation of China 81922071 (Y.Z.); Zhejiang Province Science Fund for Distinguished Young Scholars LR19H310001 (Y.Z.); Key R & D Projects of Zhejiang Province 2021C03039 (Y.Z.); the National Natural Science Foundation of China Grant (32100959 to C.M.); Y.Z. is also supported by MOE Frontier Science Center for Brain Science & Brain-Machine Integration, Zhejiang University. We thank Dr. Liang Ma for his help to synthesize compound 21 used in this study. We also thank staffs of the Center of Cryo-Electron Microscopy, Zhejiang University, BL18U beamline at National Center for Protein Sciences Shanghai (NCPSS) and BL41XU beamline of Spring-8. The diffraction data collection was performed at the BL41XU of Spring-8 with the approval of the Japan Synchrotron Radiation Research Institute (JASRI) (proposal number 2019B2705).

## Author contributions

J.Q. and Y.C. designed the expression constructs, purified the ghrelin receptor complexes, and prepared the samples for data collection toward the structures. J.Q., C.M., and D-D.S. designed the Cryo-EM experiments and prepared the cryo-EM grids, collected cryo-EM images, and performed map calculations. Q.M. built and refined the cryo-EM structure. Y.C. and Z.X. developed the ghrelin receptor construct, purification, and crystallization. K.H. and Z.X. collected diffraction data. Z.X. solved and refined the crystal structures with help of J.Z. J.Q., S-Y.J., H.Z., and C.W. designed the cellular assays and analyzed results. Y.M., Y.Z., and Z.S. planned and coordinated the project, W.Y., Y.Z., and Z.S. supervised the overall project, and wrote the manuscript.

## Competing interests

The authors declare no competing interests.
