## [Peer Review File · Nature Communications]

Molecular mechanism of agonism and inverse agonism in ghrelin receptorREVIEWER COMMENTS

Reviewer #1 (Remarks to the Author):

In this manuscript, the authors reported two GHSR structures: one is a crystal structure bound to an inverse agonist PF-05190457, and the other is a cryo-EM structure of ghrelin-bound receptor in complex with an engineered Go (miniG_o) protein via the NanoBIT strategy. The structures revealed the recognition mode of inverse agonist PF-05190457 and agonist ghrelin bound to GHSR, and compared with the published antagonist-bound GHSR crystal structure, illustrated the molecular recognition landscape for all three types of functional ligands for a constitutively active GPCR. While the manuscript provides important insight in ligand recognition and activation mechanism for GHSR, I find some fundamental questions remain to be addressed and more supporting experiments are needed.

Major comments:

1. With antagonist, inverse agonist and agonist bound GHSR structure solved, can the authors provide some conclusions regarding why GSHR keeps high constitutive activity? What's the mechanism from a structural perspective? For example, GPR52 has constitutive activity because of the built-in agonist motif from ECL2 which induces GPR52's basal activation (Lin et al, Nature, 2020. <https://doi.org/10.1038/s41586-020-2019-0>).
2. Since PF-05190457 is an inverse agonist, it should show different signaling output than agonist and antagonist. In particular, mutation of the key residues identified for specific interaction with this inverse agonist should show different response compared to antagonist and agonist. Therefore, the antagonist compound21 should be measured and included in Figure 1a and Figure 2h.
3. In page 11 line 302 "Together, our ghrelin receptor". The authors mentioned that this WFF (or more broadly, aromatic) cluster also existed in NTSR1 and other GPCRs (Page 10 line 263: For instance, activation"). Do GPCRs like NTSR1 and others containing this aromatic cluster also show constitutive activation? Can the authors draw some general conclusion from this angle?

Minor comments:

The manuscript is not well written and figures need significant work to improve. Some messages are not clearly delivered, either:

1. The first sentence of Abstract: "...remains poorly understood". This is apparently not true for all GPCRs. Indeed, inverse agonist or antagonist-bound structures were discovered years before active ones for a few GPCRs (b2AR, A2aR, for example).

2. In Page 9 line 237 “The ends of TM3, TM5, TM6 and TM7 shift 2.8 Å, 2.3 Å, 13.4 Å and 2.9 Å, respectively.” Please refer the measuring object (residues or main chain atoms).
3. In page 11 line 308. “and breaking of the weak polar interaction.....” first of all, “weak” should be “weak”. And the interaction between R102 and Q120 should be either polar interaction or salt bridge, please check and confirm.
4. The authors used a lot of red arrows to show the side chain movement, but sometimes in a confusing way. For example, in Figure 2g please make the red arrow softer (or in other color) to not to interfere with the presentation of the residues.
5. In Figure 4b there are 4 red arrows, however 5 distances are labeled in the figure, please check.
6. In Figure 5c the longest red curve arrow is very confusing, please delete and compare the corresponding residues with antagonist bound structure.
7. In Figure 5b, comparison of the WFF motif to show conformational changes should be done between antagonist-bound, inverse agonist bound, and agonist bound structures, rather than only comparing this motif between the agonist bound and inverse agonist bound structure.
8. Extended Data Fig.5: please indicate the viewing angle, for example, add text “Membrane side” at the first column and “Intracellular Side” at the third column to help readers to understand the figures.
9. Please do clear labeling on Extended Data Fig.7 and Extended Data Fig.8 to indicate which ligand (state) is which. And the two colors in panel 7a is too close to differentiate.
10. In Extended Data Table 1: the β angle was 90.60. Please check if P222 might be a better space group than P21.
11. A lot of spelling and grammatical errors. Just give one example, in line 233: “This conformation change of TM7 may due to the C terminal of ghrelin interaction with the residues on the TM7”, could be corrected to: “This conformational change of TM7 may be caused by the direction interaction of C-terminal portion of ghrelin with TM7”.

Reviewer #2 (Remarks to the Author):

This manuscript reports about the molecular structures of the ghrelin receptor in its agonist (ghrelin itself) and an inverse agonist (PF-05190457) bound forms, and discusses about the mechanism of the receptor activation. The data presented are convincing and represent a significant advance in understanding the activation mechanism of the ghrelin receptor.

Major points

1, Various mutants were made to identify amino acids associated with agonist and inverse agonist activity. However, it may be possible that the activity is decreased simply because the receptor protein expression level is low. Because the constructs have the FLAG tag, so I think the authors should estimate and present the expression levels of the mutated receptor protein.

2, In the results of the cryo-EM analysis, how many residues could the ghrelin peptide be identified from the N-terminus? In the extended data figure 3, can we see the 15th residue of the ghrelin peptide?

3, Lane 149. The authors suggested that the D99 may be a key residue for the inverse agonist recognition or the receptor activation. However, isn't the expression level of the D99A mutant lower than that of other mutants? Have you confirmed the expression level of the D99A mutant?

4, Lane 285. "The inverse agonist penetrates the WFF cluster." I don't know if this is the case in other inverse agonist papers, but I think it's a good point. For example, several published papers (DOI: <https://doi.org/10.1016/j.cell.2020.08.024>, <https://doi.org/10.1038/s42003-019-0482-7>, and <https://doi.org/10.1038/nature25758>) study about inverse agonists, and their structures have been solved. I think these ligands have penetrated deeper than W6.48. What is the relationship between these other inverse agonists and WFF? I think it's good that you present a comparison chart.

Minor points

1, In "Agonist binding pocket of ghrelin receptor" section. Is there no interaction of ghrelin with D99 or R102?

2, Lane 251. Please include the number of sites where F279 interacts with ghrelin.

3, Author contributions section. Contribution by K.H is missing.

3, Lane 264. "NTSR1 which belongs to the ghrelin family" It should be better to use the ghrelin receptor family, not the ghrelin family.

4, Lane 364. It says ghrelin is shown as aquamarine and its receptor is shown as deep salmon, but isn't it the other way around?

5, Figure 2c. The positions of the labels for Diazaspiro core and Arm-2 are not appropriate. These are positioned at the junctions of molecules.

Overall, the manuscript is interesting, and the data support the conclusions by the authors.

Reviewer #3 (Remarks to the Author):

The GHS receptor (GHSR) is a very interesting molecule and pharmacological target. It is characterized by a ~50% basal activity that provides continuous signal for food intake representing a serious problem considering the increasing number of cases of obesity in the developed countries. Pharmacological intervention could use inverse agonists that inhibit the signaling of the receptor. However, to this end, basic research and structural data are of utmost importance for further developments. The current manuscript adds significant data towards this goal and presents x-ray and cryo-EM structures of the GHS receptor with an inverse agonist and of the ghrelin/GHSR complex in the presence of Go-protein, both representing important achievements in the field. I find the result quite appealing that the lack of interaction of the inverse agonist with cavity II may be responsible for the downregulation of the basal activity of GHSR. This may be related to the lipid modification of ghrelin vs. PF-05190457. Clearly, PF-05190457 does not have such a lipid sidechain and there is no obvious structural feature that resembles a lipid chain. However, this modification is essential for the activation of GHSR by ghrelin. This is a clear and important message from the current results. The experimental procedures more or less follow the well-established protocols in the field and appear to have been done with care. The current manuscript goes beyond what is known about the structure of the GHSR. However, the structure of the GHSR in complex with Gq protein just came out in August, therefore, the manuscript needs some updating along with addressing the following issues/questions.

I would appreciate if the authors comment on the role of the hexadecyl lipid modification of ghrelin at Ser3 in the complex with Go. The authors have seen the sidechain of ghrelin in the cryo-EM map, but I would appreciate if some more information was given on the structural heterogeneity of the sidechain in the EM map. This should be done in comparison with dynamics data available from NMR and molecular modeling (Bender et al., *Structure*, 2019). Also the backbone fold of the agonist should be compared with the available NMR data and structural model.

On page 7, the authors write: "Those observations, especially for the rotamer change of R2836.55, reveal that rearrangement of the polar network in active receptor appears to tether those key residues on the TM3, TM5, TM6 and TM7, contracting agonist binding pocket of receptor. In agreement with our structural comparison, alanine substitution of the residue E1243.33, N3057.35 in receptor remarkably

reduced the activation potency induced by ghrelin while alanine substitution of the residue R2836.55 almost abolished the efficacy (Fig. 3d), suggesting that polar interactions involved in orthosteric ligand binding site play significant roles in receptor activation.” The authors could further elaborate that statement. What kind of “polar interactions” are meant?

Structural dynamics of GPCRs is an important issue and some solid-state NMR data is also available for GHSR (Schrottke et al., *Sci. Rep.*, 2017). Could the authors also give the cryo-EM perspective of the dynamics of GHSR in this connection?

As mentioned above, the authors also need to discuss and compare their results to the recently published GHSR/Gq complex data by Wang et al., *Nature Comm.*, 2021! Especially the proposed activation mechanism should be discussed in comparison to that other data.

English language usage could be improved over the entire manuscript (use articles, correct typos etc.).

Point-to-point response letter

We thank the referees for their valuable time in reviewing our manuscript and the constructive suggestions that they have provided. Please find our responses to the specific comments raised by the reviewers below. We have copied each comment in ***Black Italic***, which is followed by our own point-by-point response in ***Blue***, including details about the corresponding changes to the manuscript.

Reviewer #1:

In this manuscript, the authors reported two GHSR structures: one is a crystal structure bound to an inverse agonist PF-05190457, and the other is a cryo-EM structure of ghrelin-bound receptor in complex with an engineered Go (miniGo) protein via the NanoBIT strategy. The structures revealed the recognition mode of inverse agonist PF-05190457 and agonist ghrelin bound to GHSR, and compared with the published antagonist-bound GHSR crystal structure, illustrated the molecular recognition landscape for all three types of functional ligands for a constitutively active GPCR. While the manuscript provides important insight in ligand recognition and activation mechanism for GHSR, I find some fundamental questions remain to be addressed and more supporting experiments are needed.

Reply: We thank the referee for his/her comprehensive summary and pivotal suggestions for our work.

Major comments:

1. With antagonist, inverse agonist and agonist bound GHSR structure solved, can the authors provide some conclusions regarding why GSHR keeps high constitutive activity? What's the mechanism from a structural perspective? For example, GPR52 has constitutive activity because of the built-in agonist motif from ECL2 which induces GPR52's basal activation (Lin et al, Nature, 2020. <https://doi.org/10.1038/s41586-020-2019-0>).

Reply: We thank the referee for the insightful comment. To better understand the constitutive activity of ghrelin receptor, we analyzed different conformational states of ghrelin receptor, and compared those to the receptor that was previously described with high basal activity, like GPR52.

In brief, GPR52 contains 22 residues in ECL2 region that fold into a small module and occupies orthosteric binding pocket of the receptor (***Fig. R1a***). The segment of ECL2 region, particularly for Y185^{ECL2} and H186^{ECL2}, behaves as a built-in 'agonist' for activating GPR52, thus resulting in high level of basal activity of the receptor ¹ (***Fig. R1b***). The ECL2 of ghrelin receptor also contribute to the constitutive activity as previously reported ². By detailed comparison with the structures of GPR52, the side chain of Q^{3,29} in ghrelin receptor (equivalent to G^{3,29} in GPR52) presumably causes steric hindrance for insertion of ECL2 into the orthostatic binding pocket of ghrelin receptor (***Fig.***

R1c).

Comparing three states of ghrelin receptor in complex with antagonist, agonist and inverse agonist respectively, reveal two key features of the receptor contributing to high level of constitutive activity (**Fig. R1d**). One is the direct contact between E124^{3.33} and R283^{6.55} that links TM3 and TM6 at the upper side of the orthosteric binding pocket. The E124^{3.33}-R283^{6.55} motif could stabilize the conformation of TM6 and allow an opening cavity at the intracellular end of TM6 with respect to G protein coupling. Moreover, the salt bridge of E124^{3.33}-R283^{6.55} is broken upon inverse agonist binding. The second feature, as discussed in our revised manuscript, is observed within the WFF motif which locates at the bottom side of the hydrophobic switch core of ghrelin receptor. Distinct rotameric states of WFF motif further induce receptor transition to initial signaling cascade.

In general, comparing ghrelin receptor with GPR52 indicating that the constitutive activity of ghrelin receptor may be accomplished by the two features, E-R motif and WFF cluster, together with downstream cascade motifs like PIF motif. The notable replacements of the two key motifs in ghrelin receptor is also in consistent with previous studies that have shown dynamic characters of GPCRs facilitating divergent functions, including ligand-dependent potency and high basal activity. More information and discussion are included in the revised manuscript (Line 316).

Figure R1. Comparison of the constitutive activity of the ghrelin receptor to GPR52.

Main view (a) and top view (b) of the active state of the GPR52 structure; (c) Conformation of ECL2 region of the Ghrelin receptor; (d) E-R motif and WFF cluster synergistically stabilize the tight contract interaction between TM6 and TM3/7. (**Extended data Fig. 10**)

2. Since PF-05190457 is an inverse agonist, it should show different signaling output than agonist and antagonist. In particular, mutation of the key residues identified for specific interaction with this inverse agonist should show different response compared to antagonist and agonist. Therefore, the antagonist compound21 should be measured and included in Figure 1a and Figure 2h.

Reply: We thank the referee's insightful comment. We have performed and combined the signaling assay about the antagonist (compound 21) and agonist (ghrelin peptide) to the ghrelin receptor. The data is added in the revised manuscript and corresponding figures (see also Figure R2).

Figure R2. Signaling assay for the PF-05190457, ghrelin and compound21.

- Dose-dependent responses of wild-type ghrelin receptor to endogenous agonist ghrelin, inverse agonist PF-05190457 and antagonist compound 21 measured by cellular IP1 accumulation assays. (**Figure 1a**)
- IP1 accumulation measurement induced by inverse agonist PF-05190457(left and blue column) or antagonist compound 21 (right and red column). (**Figure 2h**)
- IP1 accumulation measurement induced by inverse agonist PF-05190457(left), agonist ghrelin (middle) and antagonist compound 21 (right). (**Extended data Fig. 6**)

3. In page 11 line 302 "Together, our ghrelin receptor". The authors mentioned that this WFF (or more broadly, aromatic) cluster also existed in NTSR1 and other GPCRs (Page 10 line 263: For instance, activation"). Do GPCRs like NTSR1 and others containing this aromatic cluster also show constitutive activation? Can the authors draw some general conclusion from this angle?

Reply: We thank the referee's valuable suggestion. In our revised manuscript, we compared some of GPCRs with reporting constitutive activity and found that these receptors contain similar aromatic clusters located in the bottom of orthosteric binding pocket.

Three GPCRs: NTSR1, NTSR2 and GPR39 were found to be constitutively active³. Sequence alignment of those receptors together with GPR52 reveals high sequence identity at the E-R motif and the WFF cluster, except GPR52 (**Fig-R3a**).

Neurotensin receptors NTSR1 and NTSR2 contain the equivalent aromatic motif W^{6.48}-Y^{6.51}-F^{7.42} but different ER motif (D/E^{3.33} in NTSR1/2, respectively). NTSR2 is with 50% of maximal activity in the absence of its peptide ligand, consistent with the ~50% basal activity of ghrelin receptor³. NTSR1, however, acquired constitutive activity via an F358^{7.42}A mutation, whereas the native NTSR1 has weakly constitutive activity^{4,5}. Interestingly, the same mutation of F^{7.42}A in both ghrelin receptor and NTSR2 decreased the constitutive activity³.

To discover the function of the aromatic residues on WFF cluster, we construct the NTSR1 mutant and assessed the basal activity compared with wild-type NTSR1. Consistent with previous report, F^{7.42}A increased basal activity of NTSR1, and both Y^{6.51}A and Y^{6.51}F increased the constitutive activity significantly (**Figure R3b**). More importantly, mutant of D^{3.33}E in NTSR1 enhance the basal activity of the receptor as well, suggesting that the residues at position 3.33 and 6.51 in NTSR1 should be engaged in constitutive activity of receptor. Interestingly, additional mutation of F^{6.52}A in the background of Y^{6.51}F or D^{3.33}E-Y^{6.51}F NTSR1 mutant decreased the constitutive activity of receptors to the similar level of wild-type NTSR1, suggesting that subtle change of the side chain in the aromatic core leading to a large cascade effect that altering the activity of receptor in the absence of agonist.

In general, mutating the Y^{6.51} to F in the NTSR1 significantly increase the basal activity of NTSR1, indicating that the WFF cluster containing highly potency to the constitutive activity. Single mutation of F^{7.42}A revealing similar effect to the Y^{6.51}F on the basal activity suggesting the complexity of the hydrophobic interaction in the aromatic core.

a

GPCRs	3.33	6.48	6.51	6.55	7.42
GHSR	E	W	F	R	F
GPR39	E	W	N	R	F
NTSR2	E	W	Y	R	F
NTSR1	D	W	Y	R	F
GPR52	S	W	Y	F	A

b

Figure R3. Sequence alignment of ghrelin receptor family. (Extended data Fig. 11)

- (a) Alignment of the E-R motif and WFF cluster in ghrelin receptor family and GPR52.
 (b) Constitutive activity of wild-type and mutant NTSR1 receptor.

Minor comments:

The manuscript is not well written and figures need significant work to improve. Some messages are not clearly delivered, either:

1. The first sentence of Abstract: "...remains poorly understood". This is apparently not true for all GPCRs. Indeed, inverse agonist or antagonist-bound structures were discovered years before active ones for a few GPCRs (b2AR, A2aR, for example).

Reply: We appreciate the referee's valuable suggestion, we have revised whole text and figures in the revised manuscript. The sentence has been modified as is shown in line 37, "...yet the molecular mechanism of such ligand remains insufficiently elucidated."

2. In Page 9 line 237 "The ends of TM3, TM5, TM6 and TM7 shift 2.8 Å, 2.3 Å, 13.4 Å and 2.9 Å,

respectively.” Please refer the measuring object (residues or main chain atoms).

Reply: We thank the referee for the suggestion. We have noted the measuring residues in the main text. “The ends of TM3, TM5, TM6 and TM7 shift 2.8 Å, 2.3 Å, 13.4 Å and 2.9 Å (comparing the C α of the residues: C146^{3.55}, L239^{5.65}, L253^{6.25} and L322^{7.52}), respectively.”

3. In page 11 line 308. “and breaking of the week polar interaction.....” first of all, “week” should be “weak”. And the interaction between R102 and Q120 should be either polar interaction or salt bridge, please check and confirm.

Reply: We thank the referee for pointing the mistakes. We corrected the word and modified the sentence in the revised manuscript, “... and breaking of the weak polar interaction between R102^{2.63} and Q120^{3.29}, newly establishing of polar interactions between inverse agonist with S308^{7.38}, and with D99^{2.60}.”

4. The authors used a lot of red arrows to show the side chain movement, but sometimes in a confusing way. For example, in Figure 2g please make the red arrow softer (or in other color) to not to interfere with the presentation of the residues.

Reply: We have modified the figures according to the referee’s suggestions.

5. In Figure 4b there are 4 red arrows, however 5 distances are labeled in the figure, please check.

Reply: We have labeled the distances in the Figure 4b.

6. In Figure 5c the longest red curve arrow is very confusing, please delete and compare the corresponding residues with antagonist bound structure.

Reply: Thanks for the referee’s suggestion. We have removed the curve arrow, and compared the corresponding residues and labeled in the new Figure 5c.

7. In Figure 5b, comparison of the WFF motif to show conformational changes should be done between antagonist-bound, inverse agonist bound, and agonist bound structures, rather than only comparing this motif between the agonist bound and inverse agonist bound structure.

Reply: Thanks for the referee’s suggestion. The structural superposition of the three ghrelin receptor structures has shown as in Figure 5b.

8. Extended Data Fig.5: please indicate the viewing angle, for example, add text “Membrane side” at the first column and “Intracellular Side” at the third column to help readers to understand the figures.

Reply: We have added the viewing angle in the revised manuscript.

9. Please do clear labeling on Extended Data Fig.7 and Extended Data Fig.8 to indicate which ligand (state) is which. And the two colors in panel 7a is too close to differentiate.

Reply: We have revised the Extended Data figures and corrected the labels in the figures as well as corresponding figure legends.

10. In Extended Data Table 1: the β angle was 90.60. Please check if P222 might be a better space group than P21.

Reply: We thank the referee's insightful suggestion. During the data processing, we used two different programs to assess the possible space group of the crystal data, the POINTLESS program in the CCP4 (**Fig. R4a**) and the Xtriage program in the PHENIX (**Fig. R4b**). Both of the program reveals that the space group of the data should be P2₁.

Even though, we noticed that two amorphous symmetric molecules present in the unit cell P2₁, each of them contains one ghrelin receptor and one fusion protein bRIL. It seems that one of the molecule could fit to another one by two steps of screw operation, which would result in higher point group P222. To explore this possibility, we superposed the two molecules and found that the conformation of each receptor is identical, yet the bRIL show discrepancy (**Fig. R4c**). Based on those observations above, we suppose that point group of P2₁ is more suitable than P222.

Figure R4. Assessing the space group of the crystal data of ghrelin receptor.

- (a-b) Brief report of the POINTLESS program in CCP4 (a), and the Xtraige program in PHENIX (b).
(c) Superposition of the two molecules in the $P2_1$ crystal structure data.

11. A lot of spelling and grammatical errors. Just give one example, in line 233: “This conformation change of TM7 may due to the C terminal of ghrelin interaction with the residues on the TM7”, could be corrected to: “This conformational change of TM7 may be caused by the direction interaction of C-terminal portion of ghrelin with TM7”.

Reply: Thanks for pointing out spelling and grammatical errors. We have polished the whole text through Springer Nature Author Services.

Reviewer #2 (Remarks to the Author):

This manuscript reports about the molecular structures of the ghrelin receptor in its agonist (ghrelin itself) and an inverse agonist (PF-05190457) bound forms, and discusses about the mechanism of the receptor activation. The data presented are convincing and represent a significant advance in understanding the activation mechanism of the ghrelin receptor.

Reply: We thank the referee for his/her positive comments.

Major points

1, Various mutants were made to identify amino acids associated with agonist and inverse agonist activity. However, it may be possible that the activity is decreased simply because the receptor protein expression level is low. Because the constructs have the FLAG tag, so I think the authors should estimate and present the expression levels of the mutated receptor protein.

Reply: We thank the referee's insightful comments. According to the referee's suggestion, we performed ELISA assays by trapping the FLAG tag to measure the expression level of various receptor mutants. The result shows the expression level of the mutants are diverse in a range between 0.5 to 2 fold with that of wild type receptor. We have normalized the maximum effects of each mutant in the related signal assays by the result of expression level assay. The basal activity assay is also normalized according to the expression level of each mutant. Detailed discussion and figures are provided in the revised manuscript. (Line 142,154 and 710, Extended data Fig. 6a and Figure R5).

2, In the results of the cryo-EM analysis, how many residues could the ghrelin peptide be identified from the N-terminus? In the extended data figure 3, can we see the 15th residue of the ghrelin peptide?

Reply: We thank the referee for pointing out the identified residues of ghrelin peptide in cryo-EM analysis. We used the 28 aa full-length ghrelin peptide, and the residues 1 to 15 of the ghrelin peptide are visible in our density map (Extended data Fig. 3c).

3, Lane 149. The authors suggested that the D99 may be a key residue for the inverse agonist recognition or the receptor activation. However, isn't the expression level of the D99A mutant lower than that of other mutants? Have you confirmed the expression level of the D99A mutant?

Reply: We thank the referee's concern about expression level of mutants of ghrelin receptor again. The expression level of D99A, F119A, R283A and F309A mutant of ghrelin receptors decreased approximately 50% than that of wild-type receptor, and the W276A and F279A mutation increased 2 fold of the expression level (**Figure R5**). We rephrased the IP accumulation assays by normalizing the transfection level to balance the expression of mutant receptors in HEK293 cells. The revised result shows that D99 is essential for the inverse agonist recognition (**Fig. 2h**). The expression levels of each mutant receptor are measured by ELISA assays, which were carried out by three independent

experiments and all experiments were repeated at least three times.

Figure R5. Expression level of ghrelin receptor mutants.

4, Lane 285. "The inverse agonist penetrates the WFF cluster." I don't know if this is the case in other inverse agonist papers, but I think it's a good point. For example, several published papers (DOI: <https://doi.org/10.1016/j.cell.2020.08.024>, <https://doi.org/10.1038/s42003-019-0482-7>, and <https://doi.org/10.1038/nature25758>) study about inverse agonists, and their structures have been solved. I think these ligands have penetrated deeper than W6.48. What is the relationship between these other inverse agonists and WFF? I think it's good that you present a comparison chart.

Reply: We thank the referee for the thoughtful suggestion.

We compared the WFF cluster in other solved inverse agonist bound receptors, including 5-HT2aR, 5-HT2cR, DRD2 and ETB. Sequence alignment of those receptors display high conserved W^{6.48} and F^{6.51} residues, yet the F^{7.42} is not conserved (**Fig. R6a**).

As is shown in **Figure R6b**, inverse agonists in DRD2, 5-HT2aR, 5-HT2cR and ghrelin receptor appear to extend deeply into orthosteric pocket, forming directly hydrophobic interaction with indole ring of "switch" residue W6.48. We note that aromatic moiety of inverse agonists pack against either left or right of W6.48 in receptors mentioned above (**Figure R6b**). Methiothepin, RIT, and Risperidone are observed to bind with the hydrophobic cavity which is on the left of W6.48 in 5-HT2aR, 5-HT2cR and DRD2 respectively, forming direct interactions with W^{6.48} - F^{6.51/6.52} - F^{5.47} clusters. Whereas, PF-05190457 and IRL2500 binds to another hydrophobic cavity which is on the right of W^{6.48} in ghrelin receptor and ETB receptor. Compared with activated 5-HT2cR and DRD2 structure, the hydrophobic W^{6.48} - F^{6.51/6.52} - F^{5.47} cluster also notable displacement in structures of inverse agonist bound receptor (**Figure R6d-f**). The deep binding conformation of inverse agonist impeded the side chain rotations of W^{6.48} and F^{6.44}, restricting the conformational change of TM6 from inactivate to the active state.

In addition, sequence alignment and structural comparison of ghrelin receptor with 5-HT2aR,

5-HT2cR, DRD2 and ETB receptor reveals that E^{3.33}-R^{6.55} motif is not conserved (**Fig R6a and c**), and the distance between TM3 and TM6 appears to play a role in ligand selectivity. Collectively, in ghrelin receptor family (including ghrelin receptor, NTSR1, NTSR2 and GPR39), the conserved E^{3.33}-R^{6.55} motif together with WFF cluster are suggested to be essential for inverse agonist binding as well as agonist activation.

Figure R6. Superposition of the constitutive activity containing receptors

- (a) Sequential alignment of the E^{3.33}-R^{6.55} motif and aromatic cluster for Ghrelin receptor, DRD2, ETB and 5-HT2a/2cR.
- (b) Comparison at the portion of WFF cluster for the receptors, the inverse agonists in the receptors except PF-05190457 are colored in gray.
- (c) Comparison at the portion of E^{3.33}-R^{6.55} motif for the receptors.
- (d-f) Comparison between the agonist bound and inverse agonist bound ghrelin receptor structures (d), 5-HT2cR structures (e) and DRD2 structures (f).

Minor points

1, In “Agonist binding pocket of ghrelin receptor” section. Is there no interaction of ghrelin with D99 or R102?

Reply: We appreciate the referee’s helpful consideration. In the ghrelin bound ghrelin receptor structure, 4.2-4.5 Å distance between side chain of R102^{2.63} and ghrelin peptide reveals a weak polar contact or salt bridging. The side chain of D99^{2.60} is not observed forming direct interaction with ghrelin receptor, but keeps interactions with Y313^{7.43} and R102^{2.63}, stabilizing active conformation of ghrelin receptor. More importantly, water-mediated interactions were reported to play a central role in activation of GPCRs based on molecule dynamic simulations ⁶ or high-resolution structure of GPCRs. The residues D99^{2.60} and R102^{2.63} may be involved in water-mediated interactions with ghrelin peptide. Taken together, mutants of those mutants decreased potency of ghrelin induced activation.

2, Lane 251. Please include the number of sites where F279 interacts with ghrelin.

Reply: In the revised manuscript, we have provided the specific residues of ghrelin which are involved in interaction with F279^{6.51}.

3, Author contributions section. Contribution by K.H is missing.

Reply: We thank the referee’s pointing out the mistake. We have modified the contribution sections in the revised manuscript.

4, Lane 264. “NTSRI which belongs to the ghrelin family” It should be better to use the ghrelin receptor family, not the ghrelin family.

Reply: We have corrected the sentence in the revised manuscript.

5, Lane 364. It says ghrelin is shown as aquamarine and its receptor is shown as deep salmon, but isn't it the other way around?

Reply: We have revised the sentence as is shown, “Ghrelin peptide and receptor are shown as deep salmon sphere and aquamarine cartoon respectively.”

6, Figure 2c. The positions of the labels for Diazaspiro core and Arm-2 are not appropriate. These are positioned at the junctions of molecules.

Reply: We thank the referee’s helpful suggestion, we have revised the labels as is shown in new Fig. 2c in the revised manuscript.

Overall, the manuscript is interesting, and the data support the conclusions by the authors.

Reply: We appreciated the referee's positive comments and insightful suggestions again.

Reviewer #3 (Remarks to the Author):

The GHS receptor (GHSR) is a very interesting molecule and pharmacological target. It is characterized by a ~50% basal activity that provides continuous signal for food intake representing a serious problem considering the increasing number of cases of obesity in the developed countries. Pharmacological intervention could use inverse agonists that inhibit the signaling of the receptor. However, to this end, basic research and structural data are of utmost importance for further developments. The current manuscript adds significant data towards this goal and presents x-ray and cryo-EM structures of the GHS receptor with an inverse agonist and of the ghrelin/GHSR complex in the presence of Go-protein, both representing important achievements in the field. I find the result quite appealing that the lack of interaction of the inverse agonist with cavity II may be responsible for the downregulation of the basal activity of GHSR. This may be related to the lipid modification of ghrelin vs. PF-05190457. Clearly, PF-05190457 does not have such a lipid sidechain and there is no obvious structural feature that resembles a lipid chain. However, this modification is essential for the activation of GHSR by ghrelin. This is a clear and important message from the current results. The experimental procedures more or less follow the well-established protocols in the field and appear to have been done with care. The current manuscript goes beyond what is known about the structure of the GHSR. However, the structure of the GHSR in complex with Gq protein just came out in August, therefore, the manuscript needs some updating along with addressing the following issues/questions.

I would appreciate if the authors comment on the role of the hexadecyl lipid modification of ghrelin at Ser3 in the complex with Go. The authors have seen the sidechain of ghrelin in the cryo-EM map, but I would appreciate if some more information was given on the structural heterogeneity of the sidechain in the EM map. This should be done in comparison with dynamics data available from NMR and molecular modeling (Bender et al., Structure, 2019). Also the backbone fold of the agonist should be compared with the available NMR data and structural model.

Reply: We are so grateful for the referee's sufficient advising that providing us to explore the role of the hexadecyl lipid modification of ghrelin.

By comparing the ghrelin peptide in the two EM complex structures (Ghrelin bound Gi complex in this study and Gq complex ⁷) and the NMR structure (PMID: 25803439), we found that ghrelin can adopt various conformations in different states. The N terminus of the ghrelin show similar conformations, except the octanoyl group on Ser⁺³ (**Figure R7a**). The octanoyl group of the ghrelin in the receptor-bound states exhibit highly dynamic feature ⁸. The conformation of the octanoyl group in the two cryo-EM structures display various orientation of the terminal methyl group locating on the different sides of TM5 helix, respectively (**Figure R7b**). Consequently, the

side chains of L210, V212 and M213 in the two ghrelin receptor structures forms different hydrophobic pockets accommodating the tail of the octanoyl group (**Figure R7c**).

We speculate that the different orientation of the octanoyl group may due to the dynamics of residues on TM5. The G-protein free state of the ghrelin receptor displaying more flexibility at the extracellular portion ⁹. Coupling with G-protein may stabilize the conformation of TM helices and octanoyl group of ghrelin may also stable in the cavity II of the orthosteric binding pocket and enhance the active form of ghrelin receptor.

The C terminus of ghrelin peptide is identical in the two cyro-EM complex structures, which form a well-defined α helix supported by the ECL3 region of ghrelin receptor. However, the C terminus of peptide encompassing residues from Glu⁺⁸ to Ser⁺¹⁸ showed more flexible and disordered in the NMR structure. Indicating that the C-terminal region is un-stable in the free state of ghrelin receptor.

Figure R7. Dynamic of the ghrelin peptide bound with ghrelin receptor

- Superposition of the ghrelin peptide in the Gq/Go coupled receptor complex structure and solution NMR structure.
- Comparison of the conformation of the ghrelin peptide at the N terminal, including octanoyl modification on Ser⁺³.
- View of the different binding model of the octanoyl groups between Gq and Go coupled receptor complex structures.

On page 7, the authors write: “Those observations, especially for the rotamer change of R283^{6.55}, reveal that rearrangement of the polar network in active receptor appears to tether those key residues on the TM3, TM5, TM6 and TM7, contracting agonist binding pocket of receptor. In agreement with our structural comparison, alanine substitution of the residue E124^{3.33}, N305^{7.35} in receptor remarkably reduced the activation potency induced by ghrelin while alanine substitution of the residue R283^{6.55} almost abolished the efficacy (Fig. 3d), suggesting that polar interactions involved in orthosteric ligand binding site play significant roles in receptor activation.” The authors could further elaborate that statement. What kind of “polar interactions” are meant?

Reply: Thank you for the question. We are sorry about the confusing sentence. We have rephrased

this section as follow:

Line 190:

These observations, particularly for the rotamer change of R283^{6.55}, reveal that rearrangement of the residues in the polar network appears to tether those key residues on TM3, TM5, TM6 and TM7, contracting the agonist binding pocket of the receptor. In agreement with our structural comparison, alanine substitution of residues E124^{3.33} and N305^{7.35} in the receptor remarkably reduced the activation potency induced by the ghrelin peptide, while alanine substitution of residue R283^{6.55} almost abolished the efficacy (Fig. 3d), suggesting that the polar interactions of E124^{3.33}, R283^{6.55}, S217^{5.43} and N305^{7.35} in the receptor and Gly⁺¹ and Ser⁺² in the ghrelin peptide play significant roles in receptor activation. In accordance with previously published Gq-coupled ghrelin receptors in complex with ghrelin structure, Gly⁺¹ and Ser⁺² in ghrelin are engaged in similar contacts with receptors (Extended data Fig. 8b).

Structural dynamics of GPCRs is an important issue and some solid-state NMR data is also available for GHSR (Schrottke et al., Sci. Rep., 2017). Could the authors also give the cryo-EM perspective of the dynamics of GHSR in this connection?

Reply: We thank the referee's constructive suggestion. GPCRs are very dynamic in cell membrane and trigger divergent signaling pathway in response to different types of ligands. Recent progress in the structural determination of GPCRs and GPCR-transducer complexes represent important steps toward deciphering GPCR signal transduction at the molecular level, however, NMR spectroscopy definitely provides a complementary approach to understand molecular dynamics of GPCRs¹⁰. Previous NMR studies have revealed that ghrelin receptor in membranes display segmental fluctuations and rapid conformational transitions on a sub-microsecond timescale, in particular for loop and terminal regions. Consistently, comparison of ghrelin bound receptor with inverse agonist bound structure reveals that extracellular regions (N-terminus, ECL2 and ECL3) and intracellular region (ICL2 and C-terminus) have different conformations (**Fig. R8**). Moreover, in the crystal structure of inverse agonist bound ghrelin receptor, part of ECL2 region could not be traced because of lacking of electron density. In addition, TM helices of ghrelin receptor behave structural plasticity when binding to different ligands.

In cryo-EM data processing, we didn't observe other notable conformations of ghrelin receptor-Go complex, one of reasons may attribute to that Go protein in the complex is nucleic acid free state, which is consistent with previously reporting that this strategy can stabilize the GPCR-G protein complex in detergent environment. Collectively, in agreement with dynamic property of β 2 adrenergic receptor, dynamics information of ghrelin receptor can contribute to a mechanistic understanding of its pharmacology.

Figure R8. Flexibility of the ECL/ICL region of ghrelin receptor bound with different ligands.

As mentioned above, the authors also need to discuss and compare their results to the recently published GHSR/Gq complex data by Wang et al., Nature Comm., 2021! Especially the proposed activation mechanism should be discussed in comparison to that other data.

Reply: We thank the referee for his or her valuable suggestion. In the revised manuscript, by comparing ghrelin-bound GHSR-Go structure with the recent reported ghrelin-bound GHSR-Gq structure, and combining with the published descriptions about ghrelin receptor activation by Wang et al. ⁷, we eventually summarized and concluded the ghrelin receptor activation mechanism as is shown blow:

- a) Gly₊₁ and Ser₊₂ of ghrelin peptide forms a polar network with S123^{3.32}, E124^{3.33}, S217^{5.43}, R283^{6.55} and N305^{7.35}, contracting orthosteric volume of the receptor, which is one of hallmarks of aminergic receptors activation transition.
- b) By structural comparison of active and inactive ghrelin receptor, notable conformational change of R283^{6.55} and E124^{3.33} was observed, swing away from the receptor helical core, which subsequently induced rotameric switches of F279^{6.51}.
- c) It is worth to note that aromatic cluster WFF, W276^{6.48}, F279^{6.51} and F312^{7.42} exhibits significant displacement, which results in a cascade of relocations of highly conserved motifs such as P5.50-I(V)3.41-F6.44 motif, D(E)3.49-R3.50-Y3.51 motif and N7.49-P7.50-xx-Y7.53 motif.
- d) Both the octanoyl group of the Ser₊₃ are well-defined shown by the cryo-EM maps and occupied the cavity II of the binding pocket in the Gi- or Gq-coupled ghrelin receptor complexes. Interestingly, the octanoic tail of Ser₊₃ stretches horizontally toward the trench between TM5 and TM6 in our resolved structure, whereas locates between TM4 and TM5 in the Gq-coupled structure (Fig. R7b,c).

Figure R7. Dynamic of the ghrelin peptide bound with ghrelin receptor

- (d) Superposition of the ghrelin peptide in the Gq/Go coupled receptor complex structure and solution NMR structure.
- (e) Comparison of the conformation of the ghrelin peptide at the N terminal, including octanoyl modification on Ser⁺³.
- (f) View of the different binding model of the octanoyl groups between Gq and Go coupled receptor complex structures.

English language usage could be improved over the entire manuscript (use articles, correct typos etc.).

Reply: We thank the referee's suggestion. We have polished the manuscript and extended data by SNAS (Springer Nature Author Services)

Reference

1. Lin, X. *et al.* Structural basis of ligand recognition and self-activation of orphan GPR52. *Nature* **579**, 152-157 (2020).
2. Mokrosinski, J., Frimurer, T.M., Sivertsen, B., Schwartz, T.W. & Holst, B. Modulation of constitutive activity and signaling bias of the ghrelin receptor by conformational constraint in the second extracellular loop. *J Biol Chem* **287**, 33488-33502 (2012).
3. Holst, B. *et al.* Common structural basis for constitutive activity of the ghrelin receptor family. *J Biol Chem* **279**, 53806-53817 (2004).
4. Barroso, S., Richard, F., Nicolas-Etheve, D., Kitabgi, P. & Labbe-Jullie, C. Constitutive activation of the neurotensin receptor 1 by mutation of Phe(358) in Helix seven. *Br J Pharmacol* **135**, 997-1002 (2002).
5. Deluigi, M. *et al.* Complexes of the neurotensin receptor 1 with small-molecule ligands reveal structural determinants of full, partial, and inverse agonism. *Sci Adv* **7** (2021).
6. Venkatakrisnan, A.J. *et al.* Diverse GPCRs exhibit conserved water networks for stabilization and activation. *Proc Natl Acad Sci U S A* **116**, 3288-3293 (2019).
7. Wang, Y. *et al.* Molecular recognition of an acyl-peptide hormone and activation of ghrelin receptor. *Nat Commun* **12**, 5064 (2021).
8. Vortmeier, G. *et al.* Integrating solid-state NMR and computational modeling to investigate the structure and dynamics of membrane-associated ghrelin. *PLoS One* **10**, e0122444 (2015).
9. Schrottke, S. *et al.* Expression, Functional Characterization, and Solid-State NMR Investigation of the G Protein-Coupled GHS Receptor in Bilayer Membranes. *Sci Rep* **7**, 46128 (2017).
10. Hilger, D., Masureel, M. & Kobilka, B.K. Structure and dynamics of GPCR signaling complexes. *Nat Struct Mol Biol* **25**, 4-12 (2018).

REVIEWERS' COMMENTS

Reviewer #1 (Remarks to the Author):

The authors satisfactorily addressed my concerns.

Reviewer #2 (Remarks to the Author):

I have checked the revised manuscript by Dr. Jiao Qin et al. In the manuscript, the authors have truly addressed all comments made during the initial submission. In particular, I appreciate the new data on the comparison of the inverse agonists binding sites in the five GPCRs.

However, I would like to request the authors as a useful information to comment in the text for compound 21, a ghrelin receptor antagonist. I am sorry if I may miss the comment on compound 21. Because in our previous paper, we synthesized compound 21, which had not been on sale, and used it for the key experiments. I wonder whether the authors synthesized compound 21 by themselves or obtained from someone else.

I think this manuscript will have a possibility to be accepted by Nature Communications.

Reviewer #3 (Remarks to the Author):

The authors have satisfactorily addressed my questions and concerns. I recommend publication of this article.

Reviewer #1:

The authors satisfactorily addressed my concerns.

Reply: We appreciated the referee's positive comment about our revised manuscript.

Reviewer #2:

I have checked the revised manuscript by Dr. Jiao Qin et al. In the manuscript, the authors have truly addressed all comments made during the initial submission. In particular, I appreciate the new data on the comparison of the inverse agonists binding sites in the five GPCRs.

However, I would like to request the authors as a useful information to comment in the text for compound 21, a ghrelin receptor antagonist. I am sorry if I may miss the comment on compound 21. Because in our previous paper, we synthesized compound 21, which had not been on sale, and used it for the key experiments. I wonder whether the authors synthesized compound 21 by themselves or obtained from someone else.

I think this manuscript will have a possibility to be accepted by Nature Communications.

Reply: We thank the referee's comment. We really appreciate the contribution of researchers on compound 21. The compound 21 is absolutely fantastic ligand for ghrelin receptor, and we really need it in our key functional experiments. Unfortunately, the ligand is not commercially available, and, for the sake of time, transportation restrictions in response to covid-19 deterred us from

requesting the compound 21 abroad. Therefore, we asked for a medicinal chemist Dr. Liang Ma at Sichuan University to synthesize compound 21 and thanks his help in acknowledgements section. In addition, the detailed information of compound 21 were provided below (Figure R1).

Figure R1. Synthesis and quality control of compound 21.

(a) MC TIC spectra of compound 21. Detect channel: W2489 ChB 220nm (top, black); QDa Positive Scan (bottom, blue). (b) QDa Positive scan of the purified compound 21 from 80.00 to 600.00 Da. (c) NMR ¹H spectra (600 MHz) of the synthesized compound 21.

Reviewer #3 (Remarks to the Author):

The authors have satisfactorily addressed my questions and concerns. I recommend publication of this article.

Reply: We appreciated the referee's positive comment about our revised manuscript.